# A universal glycoenzyme biosynthesis pipeline that enables efficient cell-free remodeling of glycans

Thapakorn Jaroentomeechai[1], Yong Hyun Kwon [1], Yiwen Liu[1], Olivia Young[1], Ruchika Bhawal [2], Joshua D. Wilson[3], Mingji Li [1], Digantkumar G. Chapla[4], Kelley W. Moremen [4], Michael C. Jewett [5], Dario Mizrachi[6] & Matthew P. DeLisa [1,2] ✉

The ability to reconstitute natural glycosylation pathways or prototype entirely new ones from scratch is hampered by the limited availability of functional glycoenzymes, many of which are membrane proteins that fail to express in heterologous hosts. Here, we describe a strategy for topologically converting membrane-bound glycosyltransferases (GTs) into water soluble biocatalysts, which are expressed at high levels in the cytoplasm of living cells with retention of biological activity. We demonstrate the universality of the approach through facile production of 98 difficult-to-express GTs, predominantly of human origin, across several commonly used expression platforms. Using a subset of these water-soluble enzymes, we perform structural remodeling of both free and protein-linked glycans including those found on the monoclonal antibody therapeutic trastuzumab. Overall, our strategy for rationally redesigning GTs provides an effective and versatile biosynthetic route to large quantities of diverse, enzymatically active GTs, which should find use in structure-function studies as well as in biochemical and biomedical applications involving complex glycomolecules.

Glycosylation—the process by which carbohydrate-based compounds known as glycans are covalently attached to acceptor molecules, typically proteins and lipids—is fundamental to all life[1,2]. Following conjugation to biomolecules, glycans add an additional layer of information and play important roles in numerous biological processes[3] including cell adhesion and signaling[4,5], cell growth and development[6], and immune recognition/response[7,8], among others. Moreover, structural remodeling of protein-linked glycans can improve therapeutic properties in a number of ways such as extending activity and stability both in vitro and in vivo[9,10], modulating interactions with specific immune receptors[11], and targeting specific cells or tissues[12].

As our appreciation for the biological roles and therapeutic potential of glycans continues to grow, so too does the need for reliable, user-friendly technologies that enable their synthesis and remodeling. However, quantitative preparation of structurally defined glycans and glycoconjugates remains technically challenging and represents a critical technology gap that limits widespread access to this important biomolecule class[13]. A major reason for this difficulty is the lack of template encoding in glycan biosynthesis, which distances carbohydrate structure and function from gene sequence. Hence, unlike nucleic acids and proteins, glycans cannot be directly produced from recombinant DNA technology. Instead, glycan biosynthesis is

[1]Robert F. Smith School of Chemical and Biomolecular Engineering, Cornell University, 120 Olin Hall, Ithaca, NY 14853, USA. [2]Cornell Institute of Biotechnology, Cornell University, Ithaca, NY 14853, USA. [3]Glycobia, Inc., 33 Thornwood Drive, Suite 104, Ithaca, NY 14850, USA. [4]Complex Carbohydrate Research Center, University of Georgia, Athens, GA 30602, USA. [5]Department of Chemical and Biological Engineering, Northwestern University, 2145 Sheridan Rd Technological Institute E136, Evanston, IL 60208-3120, USA. [6]Department of Physiology & Developmental Biology, Brigham Young University, Provo, UT 84602, USA. ✉e-mail: md255@cornell.edu

controlled by the availability, abundance, and activities of gly-coenzymes, in particular, glycosyltransferases (GTs) that catalyze the formation of specific glycosidic linkages by transferring sugar mole-cules from donor substrates (e.g., nucleotide sugar or lipid-linked sugar) to hydroxyl groups of acceptor molecules[14,15] and glycosyl hydrolases that cleave glycan structures during oligosaccharide maturation[16].

GTs exhibit unique catalytic specificities for a wide range of sugar donors and acceptor substrates and generate products with distinct anomeric configurations, which helps to explain the vast structural diversity of "glycospace". In mammals alone, it is esti-mated that there are ~7000 oligosaccharide structures[17] whose gen-eration involves more than 200 GTs[18] from 45 different protein families that have been annotated in the carbohydrate-active enzymes (CAZy) database[19]. Moreover, GTs are proficient at repli-cating the diversity of naturally occurring glycans and glycoconju-gates in unnatural contexts, leading to their emergence as powerful synthetic tools for building complex glycomolecules in the labora-tory. Much of the progress in this regard exploits sugar nucleotide-dependent GTs of mammalian and bacterial origin for the synthesis of complex carbohydrates, glycoconjugates, and glycosylated nat-ural products, which are generated by functionally reconstituting artificial networks of these glycoenzymes within model cellular systems[20–23] or in cell-free, one-pot reaction systems[24–26].

These developments notwithstanding, broad access to GTs for fundamental and applied research is bottlenecked by difficulties associated with their recombinant expression. A major reason for this difficulty is that many GTs catalyze reactions at membrane interfaces (e.g., between the cytoplasm and periplasm in Gram-negative bacteria or between the cytosol and endoplasmic reticulum (ER)/Golgi orga-nelles within eukaryotes). As such, these enzymes are typically either secretory proteins or integral membrane proteins (IMPs) that need post-translational modifications (PTMs) (e.g., disulfide bonds, N-linked glycosylation) and/or specialized chaperones to achieve proper fold-ing, membrane translocation/insertion, and function. Efforts to express GTs in the absence of required PTMs or chaperones, or in the presence of single- or multi-pass transmembrane domains (TMDs) or terminal signal peptides (e.g., N-terminal export signals, C-terminal retention signals), are often met with non-functional protein aggre-gates. This is particularly pronounced for the expression of mamma-lian GTs in bacterial hosts, with successful reports often involving time- and labor-intensive searches for solubility-enhancing fusion partners and molecular chaperones, optimal host strains and culture condi-tions, and compatible detergents and denaturants for IMP solubiliza-tion and in vitro refolding from inclusion bodies, respectively[27–29].

For these reasons, functional expression of mammalian GTs in bacteria remains rare. Instead, eukaryotic cells remain the preferred host for producing recombinant glycoenzymes albeit with most stu-dies involving small-scale expression of just one or a few GTs[15]. To date, there are only a few reports of larger-scale expression campaigns involving significant numbers of GTs: one such study describes the expression of 51 human GTs as fusions to the yeast cell wall Pir proteins to enable immobilization on the surface of *Saccharomyces cerevisiae*[30] while a second study describes the expression of 339 human gly-coenzymes as fusions to a solubility-enhancing GFP domain in either mammalian cells (human embryonic kidney (HEK) 293) or baculovirus-infected insect cells[18]. Interestingly, the authors of this latter study explore the potential of *Escherichia coli* for human glycoenzyme expression but report that all GTs expressed in this host accumulate as insoluble aggregates[18]. Thus, the biosynthetic capacity and versatility of simple *E. coli* bacteria, one of the most important model organisms in biology and biotechnology[31], is yet to be unlocked for the functional expression of GTs on a large scale.

To address this gap, we describe a generalizable workflow for the efficient production of structurally diverse GTs using standard *E. coli* expression strains. At the heart of this workflow is a protein engi-neering method called SIMPLEx (solubilization of IMPs with high levels of expression)[32] that enables the topological conversion of secretory and membrane-bound proteins into water-soluble variants. Here, this conversion is achieved for GTs by modifying their N-termini with a decoy protein that prevents membrane insertion and their C-termini with an amphipathic protein that effectively shields hydrophobic surfaces from the aqueous environment (Fig. 1a). Using this approach, we demonstrate soluble expression of nearly 100 GTs, including many of human origin, directly within the *E. coli* cytoplasm at titers in the 5–10 mg/L range. Importantly, this large-scale expression platform furnishes functional glycoenzymes that can subsequently be used to remodel the structures of diverse glycan acceptors, leading to the formation of a variety of important glycoforms including human complex-type N-glycans on the therapeutic monoclonal antibody (mAb) trastuzumab. We anticipate that SIMPLEx-remodeled GTs will help to deepen our understanding of glycoenzymes from all kingdoms of life and accelerate the assembly of these enzymes into cell-based and cell-free systems that enable the biosynthesis of important glycomolecules.

## Results
### SIMPLEx promotes soluble expression of human ST6Gal1
Toward our goal of developing a versatile and universal approach for large-scale GT production, we hypothesized that SIMPLEx could relieve bottlenecks that have hampered GT expression in *E. coli*. The rationale for this hypothesis was based on two observations. First, the SIMPLEx strategy has previously been shown as a promising technique for converting IMPs into water-soluble proteins with retention of bio-logical function[32,33]. Second, SIMPLEx was able to rescue soluble expression of a diverse panel of globular proteins that were previously reported to be recalcitrant to soluble expression in *E. coli*[34] (Supple-mentary Fig. 1a). Collectively, these results highlight the capacity of SIMPLEx to shield large amounts of protein hydrophobicity that drive misfolding and aggregation and promote soluble expression of membrane and non-membrane proteins alike.

To see if the benefits of SIMPLEx could be leveraged for GT expression, we chose the human β-galactoside-α2,6-sialyltransferase 1 (*Hs*ST6Gal1), a sialyltransferase belonging to the GT29 family, as a model GT for proof-of-concept experiments. *Hs*ST6Gal1 consists of a short N-terminal cytoplasmic tail (CT), a TMD, a stem region that serves as a linker, and a large C-terminal catalytic domain that adopts a variant GT-A fold containing a seven-stranded central β-sheet flanked by α-helices (Fig. 1a)[35]. Overexpression of *Hs*ST6Gal1 has been docu-mented in several cancer cell types[36]; hence, the ability to produce preparative amounts of *Hs*ST6Gal1 could help to understand its role in cancer biology and therapy. To express this enzyme in the SIMPLEx architecture, we designed a tripartite *Hs*ST6Gal1 chimera in which its N-terminus was genetically fused to a water-soluble "decoy" protein, namely *E. coli* maltose-binding protein lacking its N-terminal signal peptide (ΔspMBP), while its C-terminus was fused to an amphipathic "shield" protein, namely truncated human apolipoprotein A1 lacking its 43-residue globular N-terminal domain (ApoAI*), yielding ΔspMBP-*Hs*ST6Gal1-ApoAI* (hereafter Sx-*Hs*ST6Gal1) (Fig. 1a). As the removal of the transmembrane anchor segment is a common practice to improve expression and solubility of mammalian GTs[18], we also generated the Sx-Δ26*Hs*ST6Gal1 variant in which 26 amino acids from the N-terminus of *Hs*ST6Gal1, comprising its CT and TMD, were genetically removed (Fig. 1a). The *Hs*ST6Gal1 enzyme contains 3 disulfide bonds in its native structure[35]. Therefore, the commercially available *E. coli* strain named SHuffle T7 Express[37], which has been engineered with a more oxidizing cytoplasmic environment and expresses a cytoplasmic version of the disulfide bond isomerase DsbC[37], was selected as an expression host to facilitate cytoplasmic disulfide bond formation. Following the expression of the two engineered chimeras in SHuffle T7 Express cells,

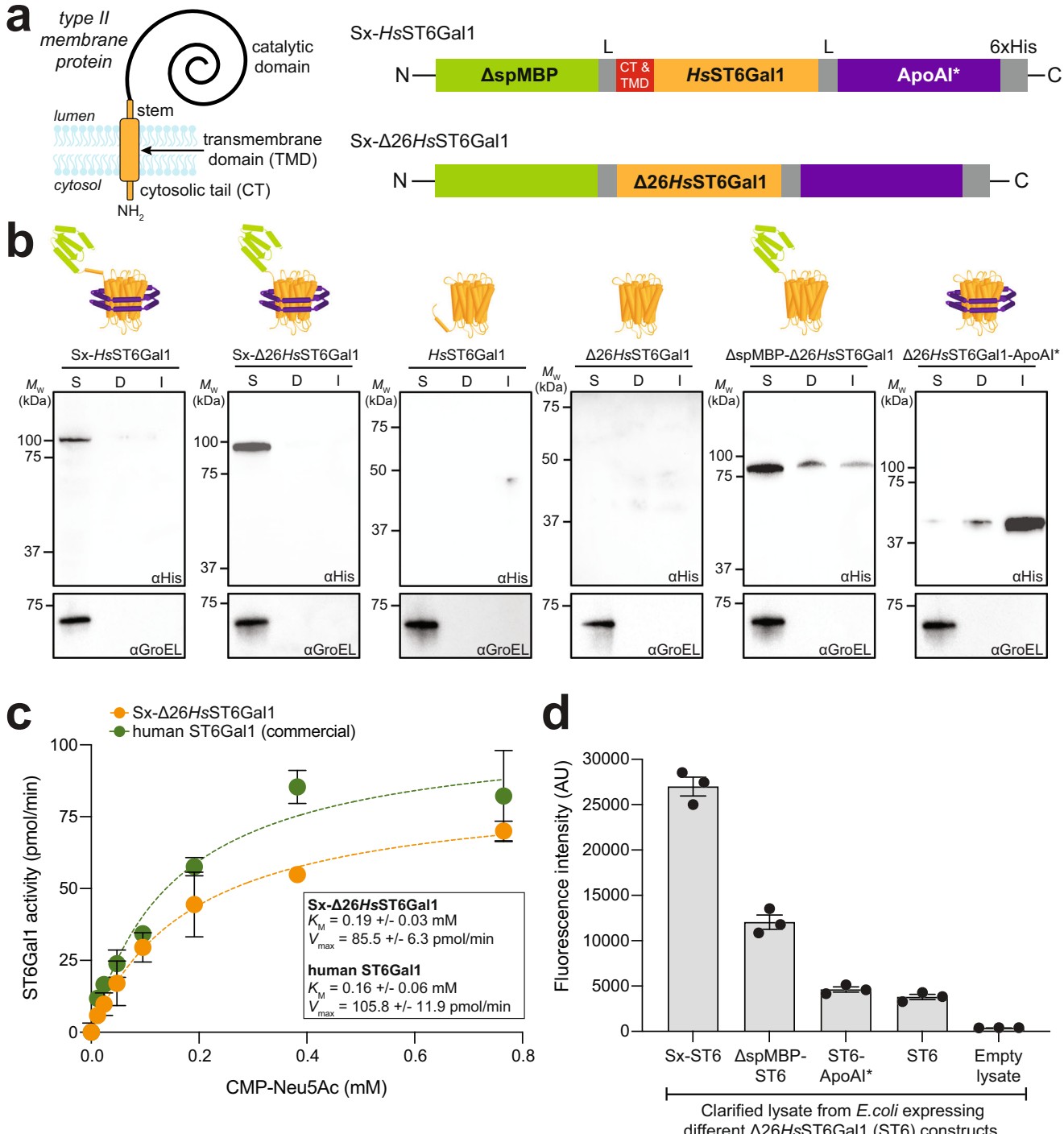

**Fig. 1 | SIMPLEx-mediated expression of biologically active *Hs*ST6Gal1.**
**a** Membrane topology of type II transmembrane proteins and molecular archi-
tecture of SIMPLEx constructs used in this study. Each construct consisted of
N-terminal ΔspMBP and C-terminal ApoAI* that flanked *Hs*ST6Gal1. Intervening
flexible linker (L) connects ΔspMBP and ApoAI* to the GT domains while the 6xHis
tag was placed at the C-terminus to facilitate detection and purification.
*Hs*ST6Gal1 domain variants studied here were wild-type (wt) *Hs*ST6Gal1 (top) and
truncated Δ26*Hs*ST6Gal1 (bottom), in which the cytoplasmic tail (CT) and
transmembrane domain (TMD) were removed. **b** Immunoblot analysis of the soluble (S),
detergent-solubilized (D), and insoluble (I) fractions prepared from *E. coli* SHuffle T7 Express
*lysY* cells carrying plasmid pET28a(+) encoding each of the indicated constructs. An
equivalent amount of total protein was loaded in each lane. Blots were probed with
anti-polyhistidine antibody (αHis). Control blots were generated by probing with
anti-GroEL antibody. Results are representative of three biological replicates.
Molecular weight ($M_w$) markers are shown at left. **c** Kinetic analysis of purified

Sx-Δ26*Hs*ST6Gal1 and commercial human ST6Gal1 performed using asialofetuin as
acceptor substrate and CMP-Neu5Ac as donor substrate. A standard phosphate
curve was generated to convert the initial raw absorbance reading to the enzyma-
tically released inorganic phosphate from CMP-Neu5Ac. Values for $V_{max}$ and $K_m$
values were determined using Prism 9. Data are the mean of three biological repli-
cates ± SEM. **d** Functional characterization of sialyltransferase-mediated che-
moenzymatic remodeling of protein-linked glycans using bioorthogonal click
chemistry-based assay. Fluorescence (501/523 nm ex/em) measured in clarified
lysates prepared from *E. coli* cells expressing: Sx-Δ26*Hs*ST6Gal1 (Sx-ST6), ΔspMBP-
Δ26*Hs*ST6Gal1 (ΔspMBP-ST6), Δ26*Hs*ST6Gal1-ApoAI (ST6-ApoAI), or Δ26*Hs*ST6Gal1
(ST6), as indicated. Lysates from *E. coli* cells carrying empty pET28a(+) plasmid were
used as a negative control (empty lysate). Fluorescence data, corresponding to the
extent of chemoenzymatic modification, are the mean of three biological replicates
(starting from freshly transformed cells) ± SEM. Source data are provided as a
Source Data file.

we observed stable products corresponding to Sx-HsST6Gal1 and Sx-Δ26HsST6Gal1 that accumulated almost exclusively in the soluble cytoplasmic fraction (Fig. 1b). In stark contrast, no detectable expression of unfused HsST6Gal1 or Δ26HsST6Gal1 was seen in the soluble fraction and only minimal amounts were observed in the insoluble and detergent-solubilized fractions (Fig. 1b), in agreement with previous findings that human sialyltransferases are poorly expressed in bacteria[27,38]. The large expression difference seen for the Sx-Δ26HsST6Gal1 fusion relative to its unfused counterpart was also clearly observed in whole cell lysates (Supplementary Fig. 1b).

To demonstrate the importance of the decoy and shield domains, we also expressed chimeras lacking each of these elements. When the decoy protein was omitted, Δ26HsST6Gal1-ApoAI* partitioned almost entirely in the insoluble fraction (Fig. 1b) and was undetectable in whole cell lysates (Supplementary Fig. 1b). Omission of the shield protein resulted in accumulation of ΔspMBP-Δ26HsST6Gal1 primarily in the soluble fraction, but with significant amounts also detected in the detergent-solubilized and insoluble fractions (Fig. 1b). Consistent with earlier studies with IMP targets[32,33], our results confirm the importance of the decoy and shield in directing folding away from the membrane and promoting water solubility. Moreover, SIMPLEx-based expression in a redox-engineered bacterial host sidestepped the need for chaperones that occur uniquely in the mammalian secretory pathway and for N-linked glycosylation of the GT that is not required for activity but needed for folding, stability, and solubility of the enzyme[39,40].

## Soluble HsST6Gal1 in the SIMPLEx framework retains biological activity

To determine whether soluble Sx-Δ26HsST6Gal1 was biologically active, the enzyme was purified (Supplementary Fig. 2a) and subjected to kinetic analysis using a commercial kit for quantifying the release of nucleotide cytidine 5'-monophosphate (CMP) from the donor substrate CMP-N-acetylneuraminic acid (CMP-Neu5Ac). From this assay, the apparent $K_M$ and $V_{max}$ values for Sx-Δ26HsST6Gal1 were determined as $0.19 \pm 0.03$ mM and $85.5 \pm 6.3$ pmol/min, respectively (Fig. 1c). These parameters were in reasonable agreement with the apparent kinetic parameters that we measured for the commercial human ST6Gal1 (produced recombinantly using N60 mouse myeloma cells) and that were measured previously[41]. The specific activity of the soluble Sx-Δ26HsST6Gal1 enzyme was 516.9 pmol/min/µg (Supplementary Fig. 2b), which was also consistent with previously published data for human ST6Gal1[42].

Upon confirming that Sx-Δ26HsST6Gal1 was enzymatically active, we next sought to demonstrate its practical utility for chemoenzymatic remodeling of N-linked glycans present on glycoprotein substrates. To this end, we developed a bioorthogonal click chemistry-based assay for quantifying sialyltransferase-mediated chemoenzymatic modification (Supplementary Fig. 2c). Specifically, Sx-Δ26HsST6Gal1 enzyme preparations were evaluated for their ability to transfer azido-Neu5Ac from CMP-activated glycosyl donor onto terminal Gal residues of the alpha-1 antitrypsin (A1AT) serpin protein, which was first treated with neuraminidase to remove native sialic acids. The modified A1AT was then fluorescently labeled through a strain-promoted azide-alkyne cycloaddition reaction using carboxyrhodamine 110 DBCO and separated by standard SDS-PAGE. Fluorescence intensity of the labeled A1AT proteins, which corresponded to the extent of chemoenzymatic remodeling by Sx-Δ26HsST6Gal1, was then directly visualized and quantified by in-gel fluorescence analysis.

Using clarified lysate generated from E. coli cells expressing Sx-Δ26HsST6Gal1 as a catalyst source, we detected strong fluorescence from the treated A1AT (Fig. 1d). In contrast, clarified lysates containing either Δ26HsST6Gal1 or Δ26HsST6Gal1-ApoAI* yielded only a weak fluorescent signal (Fig. 1d), which was consistent with the barely detectable levels of soluble expression observed for these constructs

that both lacked the ΔspMBP decoy (Fig. 1b). Interestingly, while addition of the ΔspMBP moiety alone was able to promote soluble expression of ΔspMBP-Δ26HsST6Gal1 (Fig. 1b and Supplementary Fig. 1b), the clarified lysate containing this construct exhibited about 50% less activity than that measured for the Sx-Δ26HsST6Gal1 enzyme (Fig. 1d). A significant portion of the soluble ΔspMBP-Δ26HsST6Gal1 protein lacking ApoAI* was misfolded aggregates, consistent with previous findings[32] and indicative of the essential nature of both ΔspMBP and ApoAI* domains for producing this GT in a highly soluble, active conformation within the E. coli cytoplasm. Importantly, we also confirmed that purified Sx-Δ26HsST6Gal1-catalyzed chemoenzymatic remodeling to an extent that was indistinguishable from that of commercial human ST6Gal1 (Supplementary Fig. 2d, e). The fact that the fused ApoAI* domain did not measurably interfere with important C-terminal catalytic regions in Δ26HsST6Gal1 including sialyl motifs III (Tyr354−Gln357), S (Pro321−Phe343), and VS (His370−Glu375) suggests that its helical bundle structure is sufficiently flexible to promote solubility while still allowing proper protein-glycan interactions that are necessary for native-like enzyme function.

## Large-scale soluble expression of diverse GTs using SIMPLEx platform

Encouraged by the ability of SIMPLEx to promote soluble expression of HsST6Gal1 in E. coli while preserving its biological activity, we next investigated whether the strategy could be extended to a larger collection of structurally diverse GTs. To this end, we compiled a library of 98 GT genes from diverse prokaryotic and eukaryotic organisms, with an emphasis placed on those of human origin (Supplementary Dataset 1). These genes were organized according to their species of origin and activity, and included the following: human fucosyltransferases (HsFucTs), human galactosyltransferases (HsGals), human glucosyltransferases (HsGlcTs), human mannosyltransferases (HsManTs), human N-acetylgalactosyltransferases (HsGalNAcTs), human N-acetylglucosaminyltransferases (HsGlcNAcTs), human sialyltransferases (HsSiaTs), and a collection of other human, eukaryotic, and prokaryotic GTs. Using the UniProt database[43], we annotated these GTs based on the following characteristics: (i) single-pass transmembrane protein with C-terminus in cytoplasm (type I transmembrane protein); (ii) single-pass transmembrane protein with N-terminus in cytoplasm (type II transmembrane protein); (iii) multi-pass transmembrane protein; (iv) secretory protein with N-terminal signal peptide and C-terminal ER retention domain; and (v) cytosolic protein. It is known that N-/C-terminal TMDs as well as C-terminal ER retention domains in mammalian GTs are used as membrane anchors and are dispensable for catalytic activity[44], as was seen above for HsST6Gal1. Because SIMPLEx-mediated solubility enhancement of HsST6Gal1 was independent of whether the TMD was present or absent (Fig. 1b), we generally removed these terminal TMD anchors from our designed constructs. N-terminal signal peptides that natively route GTs to the secretory pathway were not necessary in the context of our bacterial cytoplasmic expression system and thus were also removed. GTs containing internal single-pass or multi-pass TMDs as well as predicted cytosolic GTs were designed as full-length genes. Each designed construct in our GT library (see Supplementary Dataset 1 for amino acid sequences) was cloned into a T7 promoter-based expression vector as both a stand-alone GT (full-length or truncated) with C-terminal 6xHis tag (hereafter GT) and a tripartite SIMPLEx fusion (hereafter Sx-GT). The expression of all Sx-GT constructs was tested in small-scale, batch-mode microbial cultures. SHuffle T7 Express cells were used to produce enzymes containing previously observed or predicted disulfide bonds while BL21(DE3) cells were used to express enzymes without such bonds (Supplementary Dataset 1). Cytoplasmic expression of the Sx-GTs was profiled by immunoblot analysis of clarified lysates derived from E. coli cells expressing the respective constructs. Importantly, 95 of the Sx-GT constructs showed clearly visible accumulation in the

soluble cytoplasmic fractions, with most exhibiting moderate to strong expression and only a few that were faintly expressed (Fig. 2 and Supplementary Fig. 3). It should be noted that this success rate was achieved under standard bacterial expression conditions (starting $OD_{600} \approx 0.6$, induction with 0.1 mM isopropyl β-D-1-thiogalactopyranoside (IPTG) at 16 °C for 16–20 h in Luria–Bertani (LB) medium) that were identical for each construct and did not require any of the lengthy optimization trials that are commonly associated with expression campaigns employing bacteria. Conversely, only ~45% of the unfused GT constructs could be detected in the soluble fraction under the same conditions, and in most cases, the level of soluble GT expression was visibly lower compared to its Sx-GT counterpart (Fig. 2 and Supplementary Fig. 3). Subcellular fractionation analysis of 9 select candidates revealed that all SIMPLEx constructs accumulated predominantly in the soluble fraction whereas unfused versions of the enzymes partitioned mostly in the insoluble or detergent-solubilized fractions (Supplementary Fig. 4), consistent with the solubility profiles observed above for Sx-*Hs*ST6Gal1 and Sx-Δ26*Hs*ST6Gal1. In addition, aberrant expression products such as high-molecular-weight aggregates and proteolytic degradants were prevalently detected among the GT but not the Sx-GT constructs (Fig. 2), highlighting the intrinsic ability of the SIMPLEx strategy to enhance intracellular stability and prevent off-pathway misfolding and aggregation of target enzymes.

Another advantage of expressing GTs in the SIMPLEx framework is the potential to relieve cellular stress that arises from a high-level accumulation of severely misfolded proteins (e.g., inclusion bodies) or destabilization of the cytoplasmic membrane caused by high-level expression of membrane proteins, phenomena that are both well-known to negatively impact cell growth and productivity. Indeed, we consistently observed that cultures expressing Sx-GTs reached higher final cell densities than those expressing unfused GTs (Supplementary Fig. 5). Likewise, titers of selected Sx-GT candidates purified from 1-L cultures were also higher on both a mass and molar basis relative to unfused GTs, with all SIMPLEx constructs accumulating in the 5–10 mg/L range (Supplementary Fig. 6). Taken together, these results demonstrate (i) large-scale GT expression using *E. coli* as the host organism and (ii) SIMPLEx as a universal strategy for high-yield, soluble expression of GTs having diverse origins, structures, and activities.

## Correlates of successful GT expression in *E. coli*

We next sought to identify the protein features that correlated with soluble protein expression by comparing physicochemical properties of the proteins including molecular weight ($M_w$), isoelectric point (pI), and amino acid content. This involved assigning an expression score to each of the Sx-GT and GT constructs based on their soluble expression profiles (Supplementary Fig. 7a). We then used the expression scores to bin these proteins into four groups: non-expressor (score 0); weak expressor (score 1); medium expressor (score 2); and strong expressor (score 3). According to this classification, ~95% of the Sx-GTs were identified as expressible (score ≥1), with more than 50% falling into the medium-to-high expressor groups. In stark contrast, over 50% of the GT constructs were identified as non expressors, with most of the others classifying as weak expressors (Supplementary Fig. 7a). A scatter plot of protein $M_w$, excluding added mass from the ΔspMBP and ApoAI* domains, versus solubility score calculated by Protein-Sol, a web tool for predicting protein solubility from sequence[45], revealed that expressible Sx-GT constructs clustered within a 25–60 kDa range whereas expressible GT constructs were clustered in a narrower 25–40 kDa range that was skewed to smaller proteins (Supplementary Fig. 7b).

This observation prompted us to further investigate the relationship between soluble expression of the protein and its $M_w$. To this end, we categorized all GTs into one of three size groups: small ($M_w < 40$ kDa), medium ($M_w = 40–60$ kDa), and large ($M_w > 60$ kDa).

We then calculated average expression score ($\overline{Ex}$) for each size group within the Sx-GT or GT datasets. For GT constructs, a significant decrease in $\overline{Ex}$ was observed as protein $M_w$ increased, with no soluble expression for large proteins (Supplementary Fig. 8a), consistent with the observation that bacterial translation machinery has evolved to express shorter polypeptides[46] and that expression of larger eukaryotic proteins in bacteria frequently leads to misfolding and aggregation[34,47]. On the contrary, $\overline{Ex}$ was high for all Sx-GT constructs, with no significant difference between small- and medium-sized proteins and only a small decrease in $\overline{Ex}$ for large-sized proteins (Supplementary Fig. 8a). These results suggested that the SIMPLEx framework helps to overcome the protein size barrier that typically restricts successful expression in *E. coli*. Unfortunately, attempts to identify additional parameters such as protein pI and amino acid content that correlated with expressibility did not yield conclusive results (Supplementary Figs. 7b and 8b). Nonetheless, the data presented here reveal important design parameters that could guide efforts to express even more GT enzymes in the future.

## Efficient production of Sx-GTs across diverse expression platforms

To further expand the utility of the platform and demonstrate its universality, we attempted to produce SIMPLEx fusions in other popular expression platforms including (i) *E. coli*-based cell-free protein synthesis (CFPS); (ii) *Saccharomyces cerevisiae* strain SBY49; and (iii) HEK293T cells. Using an appropriate expression vector for each system, we observed a significant accumulation of the Sx-Δ26*Hs*ST6Gal1 construct in the soluble fractions derived from each of these three systems (Fig. 3). In contrast, little to no soluble expression of the unfused Δ26*Hs*ST6Gal1 construct lacking the ΔspMBP and ApoAI* domains was detected in any of these systems (Fig. 3). While Sx-Δ26*Hs*ST6Gal1 was also found in the insoluble fraction derived from the CFPS system, the amount of this construct that partitioned in the soluble fraction was significantly higher (Fig. 3a). For cell-based expression of Sx-Δ26*Hs*ST6Gal1, both yeast and human cells yielded products that accumulated almost exclusively in the soluble fractions (Fig. 3b, c), in line with the *E. coli* cell-based expression results observed above. Importantly, these results highlight the cross-platform compatibility of the SIMPLEx strategy and the ease with which it was adapted to these microbial, mammalian, and cell-free expression systems.

## Cell-free construction of free human *N*-glycans using Sx-GTs

To date, a growing number of cell-free bio/chemoenzymatic synthesis strategies have been reported that provide access to large repertoires of pure and chemically defined glycans, especially complex structures that are otherwise difficult to obtain by conventional chemical synthesis[48–50]. Because these approaches generally depend on the availability of glycoenzymes, many of which cannot be recombinantly expressed or purified at scale, we sought to demonstrate the practical utility of Sx-GTs as biocatalysts for constructing customized glycan structures via a previously described bioenzymatic synthesis approach[48]. To this end, we devised two multi-GT enzyme pathways for de novo biosynthesis of a library of human hybrid- and complex-type *N*-glycans starting from a mannose₃-*N*-acetylglucosamine₂ (Man₃-GlcNAc₂) primer (Fig. 4a). To generate this primer, we leveraged a glycoengineered *E. coli* strain carrying a heterologous biosynthesis pathway for producing undecaprenyl-linked Man₃GlcNAc₂ glycan[51]. Following glycolipid extraction from these cells, Man₃GlcNAc₂ (M3; glycan 1) was removed from undecaprenol by mild acid hydrolysis and purified to homogeneity as confirmed by matrix-assisted laser desorption/ionization-time of flight mass spectrometry (MALDI-TOF MS) analysis (Fig. 4b).

Using 1 as a primer, glycan elaboration with GlcNAc was carried out by sequential treatment with purified Sx-Δ29*Hs*GnTI and Sx-

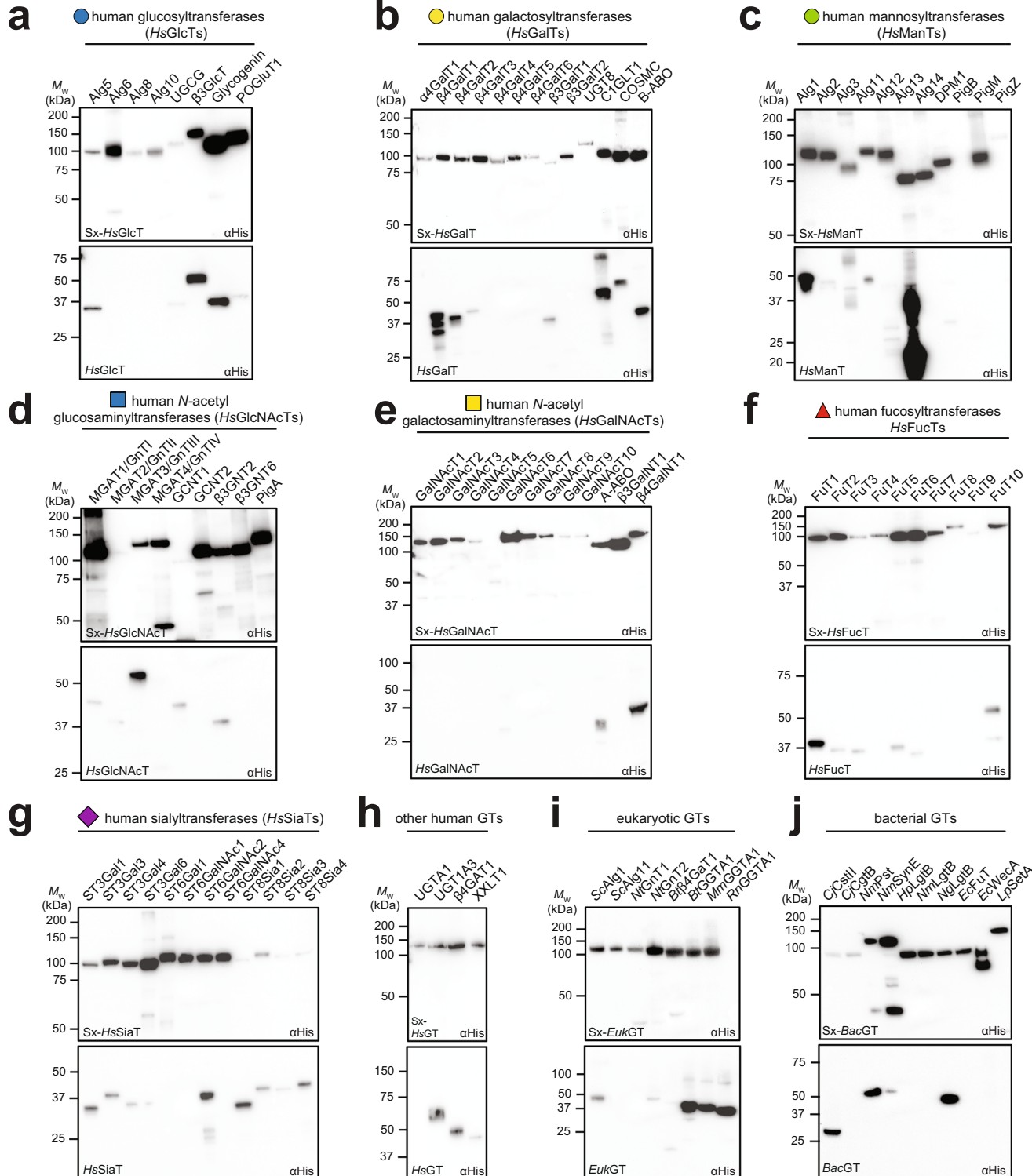

**Fig. 2 | Soluble expression of Sx-GT constructs in the *E. coli* cytoplasm.** Ninety-eight GTs were evaluated for soluble, cytoplasmic expression in the SIMPLEx framework. Immunoblot analysis of soluble fractions derived from either BL21(DE3) or SHuffle T7 Express cells carrying plasmids for Sx-GT (top blot in each panel) or unfused GT (bottom blot in each panel) constructs. GTs were clustered according to origin and activity as follows: **a** human glucosyltransferases (*Hs*GlcTs); **b** human galactosyltransferases (*Hs*GalTs); **c** human mannosyltransferases (*Hs*ManTs); **d** human *N*-acetylglucosaminyltransferases (*Hs*GlcNAcTs); **e** human *N*-acetylgalactosaminyltransferases (*Hs*GalNAcTs); **f** human fucosyltransferases (*Hs*FucTs); **g** human sialyltransferases (*Hs*SiaTs); **h** other human GTs (*Hs*GTs);

**i** eukaryotic GTs (*Euk*GTs); and **j** bacterial GTs (*Bac*GTs). The expression strain and sequence for each GT including information about truncation of TMD domains are provided in Supplementary Dataset 1. Graphical representations of monosaccharide substrates are presented according to symbol nomenclature for glycans (SNFG; https://www.ncbi.nlm.nih.gov/glycans/snfg.html). An equivalent amount of total protein was loaded in each lane and blots were probed with anti-polyhistidine antibody (αHis) to detect GTs. To confirm equivalent loading, the same samples were probed with anti-GroEL antibody (see Supplementary Fig. 3). Blots are representative of three biological replicates. Molecular weight ($M_w$) markers are indicated on left.

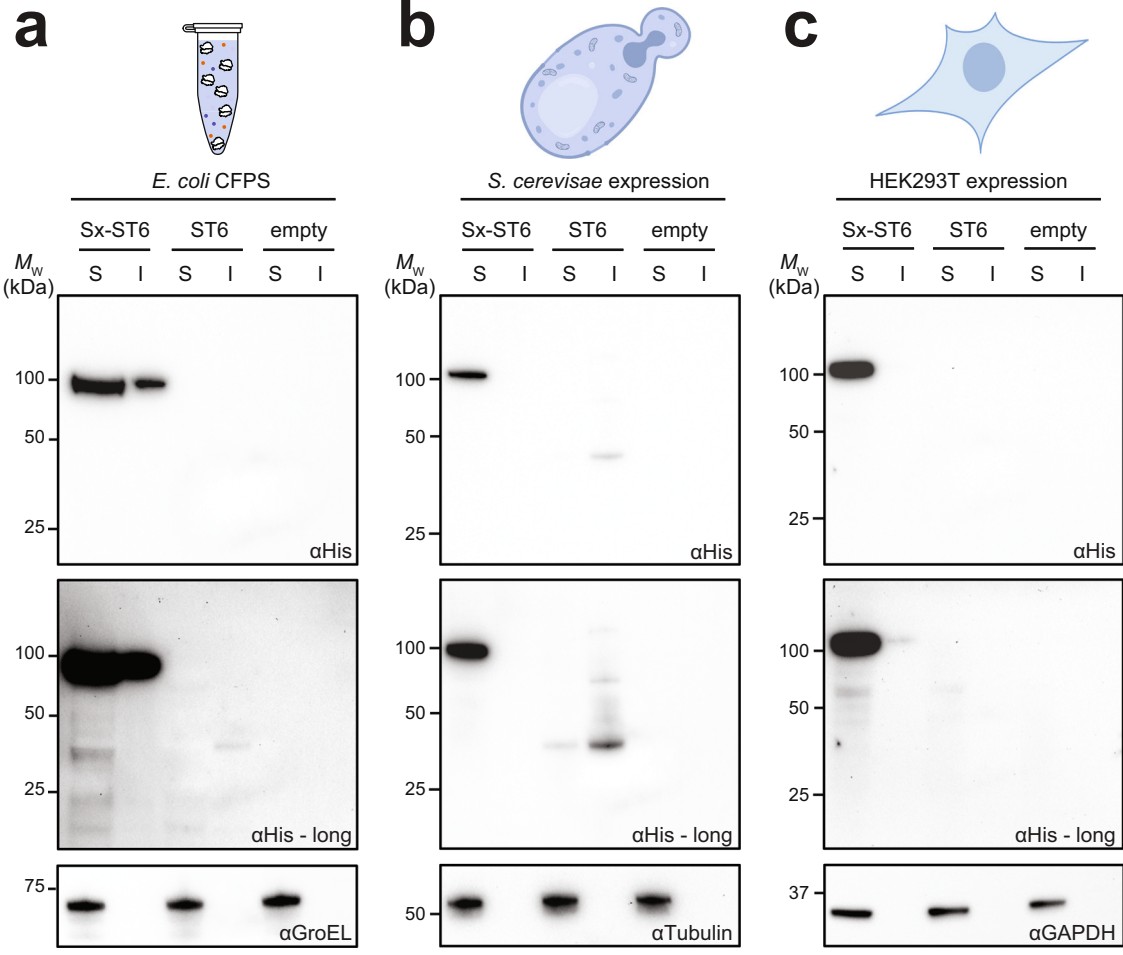

**Fig. 3 | Compatibility of SIMPLEx reformatting with diverse expression platforms.** Immunoblot analysis of the soluble (S) and insoluble (I) fractions derived from **a** cell-free protein synthesis (CFPS) using crude S30 extract prepared from *E. coli* BL21(DE3) cells; and cell-based expression using **b** *S. cerevisiae* strain SBY49 or **c** HEK293T cells as indicated. All three systems involved plasmids for expressing either Sx-Δ26*Hs*ST6Gal1 (Sx-ST6) or unfused Δ26*Hs*ST6Gal1 (ST6). Empty plasmid was used as a negative control (empty) in each case. Blots were probed with anti-

polyhistidine (αHis) antibody to detect GT expression, with longer exposures (αHis - long) provided to better identify protein products with low expression. An equivalent amount of total protein was loaded in each lane and confirmed by probing blots with antibodies specific for GroEL, Tubulin, and GAPDH, which are housekeeping proteins in *E. coli*, yeast, and mammalian cells, respectively. Results are representative of three biological replicates. Molecular weight ($M_w$) markers are shown on the left. Cartoon images were created with BioRender.com.

Δ29*Hs*GnTII, yielding hybrid-type glycan 2 (also known as G0-GlcNAc) and complex-type glycan 3 (G0), respectively, as evidenced by MALDI-TOF MS analysis of each reaction (Fig. 4b). Further elaboration of glycan 3 with galactose was achieved using Sx-Δ44*Hs*β4GalT1 to generate glycan 4 (G2), which was subsequently elaborated using Sx-Δ26*Hs*ST6Gal1 to produce glycan 5 (G2S1) and glycan 6 (G2S2), the mono- and di-sialylated complex-type *N*-glycans, respectively (Fig. 4b). Alternatively, glycan 3 was first fucosylated using Sx-Δ30*Hs*FucT8 to generate glycan 7 (G0F), which was then further elaborated to yield glycan 8 (G2F), glycan 9 (G2S1F), and glycan 10 (G2S2F) using a similar bioenzymatic strategy (Fig. 4b).

Overall, enzymatic conversion in each of these reactions was at or near 100% except in the cases involving the Sx-Δ26*Hs*ST6Gal1-catalyzed sialylation reactions. However, because the unstable nature of sialic acid-containing glycans in MALDI-TOF MS may have confounded the sialylation analysis, we performed nano-scale reverse phase chromatography and tandem MS (nano LC-MS/MS) analysis to confirm the abundance and identity of the sialylated glycans 5, 6, 9, and 10. While both mono- and di-sialylated products were clearly detected, this analysis revealed an approximate 5:1 ratio between the G2S1 and G2S2 glycans as well as the G2S1F and G2S2F glycans (Supplementary Figs. 9 and 10). It is worth pointing out that this

phenomenon has been well documented[52,53] and arises from the fact that human ST6Gal1 exhibits a preference for α1–3Man-β1,2-GlcNAc-β1,4-Gal (hereafter α1–3Man branch). As a result, ST6Gal1 readily installs Neu5Ac on this branch first, with subsequent sialylation of α1–6Man-β1,2-GlcNAc-β1,4-Gal (hereafter α1–6Man branch) known to be very slow[52].

## Cell-free remodeling of protein-linked *N*- and *O*-glycans using Sx-GTs

Glycoform manipulation is an emerging strategy for improving pharmacokinetics and pharmacodynamics of therapeutic glycoproteins[54,55]. The remodeling of protein-linked glycans can be readily achieved using one or more GTs; however, the limited availability of requisite enzymes for customizing glycan structures represents a barrier to widespread adoption. To address this technology gap, we employed members from our library of SIMPLEx-reformatted GTs to alter the glycan profiles on several biomedically relevant glycoproteins. Remodeling reactions included: (i) Sx-*Cj*CstII-mediated α2,3-sialylation of the *N*-glycoforms on α1-antitrypsin (A1AT), a serpin used in prophylactic treatment of the genetic disorder α1-antitrypsin deficiency; (ii) Sx-Δ36*Hs*FucT7-mediated fucosylation of the *N*-glycoforms on A1AT; (iii) Sx-Δ34*Hs*ST3Gal1-mediated α2,3-sialylation of the *O*-

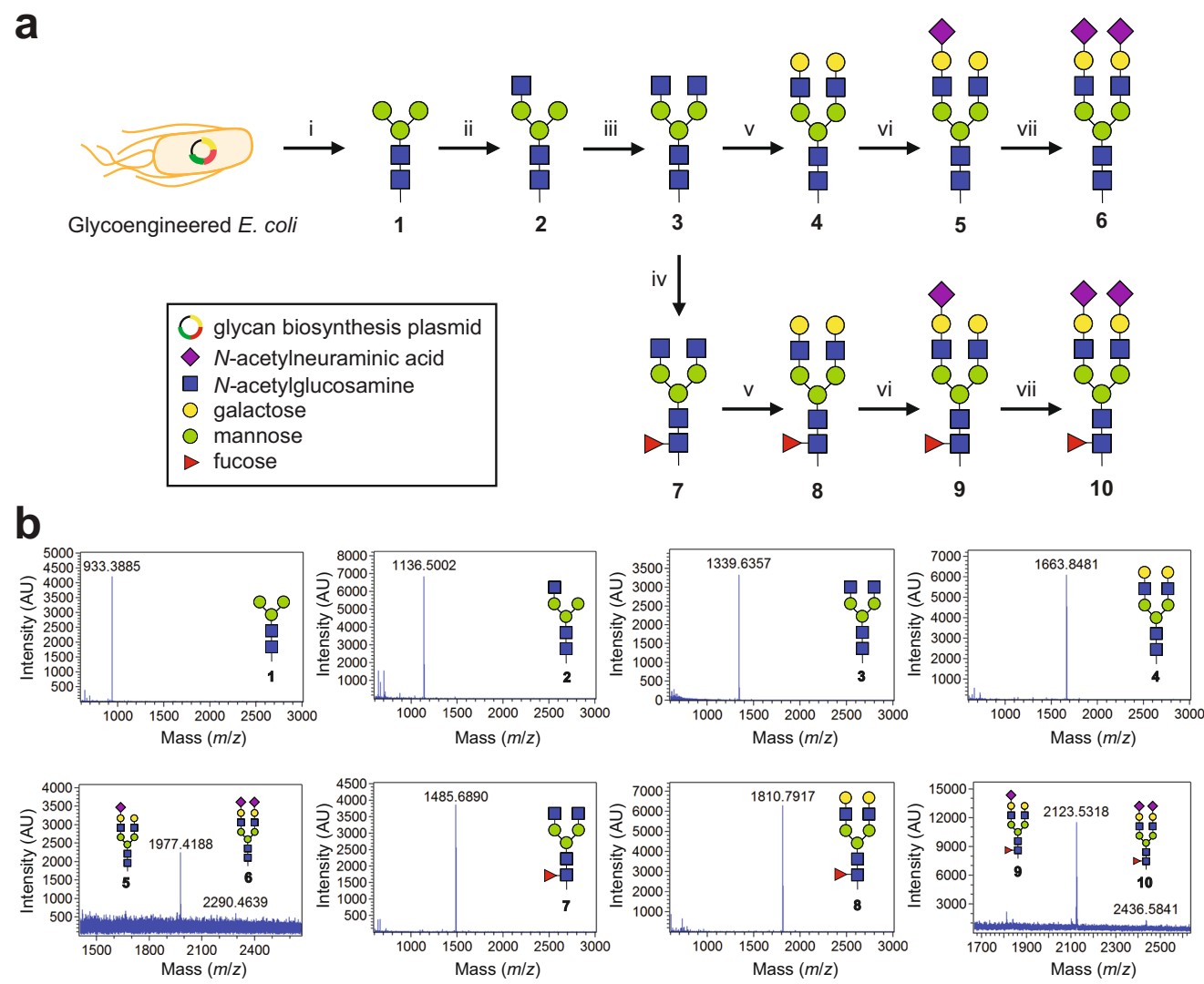

**Fig. 4 | Cell-free construction of hybrid- and complex-type *N*-glycans using Sx-GTs. a** Schematic of bioenzymatic routes to hybrid- and complex-type *N*-glycan structures. Man₃GlcNAc₂ glycan (M3; glycan 1) derived from glycoengineered *E. coli* cells equipped with biosynthesis pathway for eukaryotic trimannosyl core *N*-glycan was used as primer for glycan construction. Subsequent cell-free glycan elaboration reactions yielded the following *N*-glycan structures: 2 G0-GlcNAc; 3 G0; 4 G2; 5 G2S; 6 G2S2; 7 G0F; 8 G2F; 9 G2S1F; and 10 G2S2F. Glycan naming follows shorthand notation for IgG glycans. For a complete glycan list with chemical structures, see Supplementary Table 2. Synthesis steps: (i) non-enzymatic acid hydrolysis; (ii) Sx-Δ29*Hs*GnTI; (iii) Sx-Δ29*Hs*GnTII; (iv) Sx-Δ30*Hs*FucT8; (v) Sx-Δ44*Hs*β4GalT1; and (vi, vii) Sx-Δ26*Hs*ST6Gal1. All Sx-GTs were produced using *E. coli* BL21(DE3) or its derivative SHuffle T7 Express *lysY*. **b** MALDI-TOF MS spectra of glycans 1–10, where glycan 1 served as primer that was used as starting material to generate enzymatically derived product glycans 2–10.

glycoforms on bovine submaxillary mucin, a glycoprotein with potential uses as a biocompatible material and drug delivery vehicle; and (iv) Sx-Δ29*Hs*GnTI-catalyzed GlcNAc transfer onto Man₃GlcNAc₂ glycans present on a neoglycoprotein variant of human glucagon (GCG). In all cases, Sx-GTs readily remodeled their glycoprotein substrates, installing respective monosaccharides in 1-h reactions that were monitored using bioorthogonal click chemistry-based assays with either a fluorophore or biotin reporter for glycan labeling (Supplementary Fig. 11). It should be noted that significantly decreased activity was observed for Sx-Δ36*Hs*FucT7 when the *N*-glycans on A1AT were pre-treated with neuraminidase to remove native Neu5Ac residues. This observation was in line with earlier reports[56] and highlights how subtle differences in substrate specificity can be directly investigated using GTs within the SIMPLEx framework.

**Remodeling IgG *N*-glycans using Sx-GTs**
*N*-glycans present on the Fc domain of IgG antibodies play a critical role in the structure and function of these important proteins, but

our understanding of how discrete glycan structures affect IgG behavior remains limited due to naturally occurring microheterogeneity. Hence, strategies for generating structurally defined *N*-glycans on IgG-Fc are expected to improve our understanding of the roles played by these structures in human immunity and to open the door to creating better medicines through glycoengineering. To this end, we leveraged members from our library of Sx-GTs to generate a homogenously glycosylated variant of trastuzumab (Fig. 5a), an anti-human epidermal growth factor receptor 2 (HER2) mAb used to treat HER2-positive breast, gastroesophageal, and gastric cancers. This involved first preparing trastuzumab using a glycoengineered cell line, Expi293F™ GnTI⁻, that homogeneously produces *N*-glycoproteins bearing Man₅GlcNAc₂ glycans (Fig. 5a, glycan 11). Using a glycosidase sensitivity assay coupled with LC-MS analysis of the intact antibody, we confirmed that the *N*-glycans on trastuzumab derived from Expi293F™ GnTI⁻ were indeed Man₅GlcNAc₂ glycans (Supplementary Fig. 12). Next, Sx-Δ29*Hs*GnTI was used to install GlcNAc on the α1,3-man branch of 11 to generate

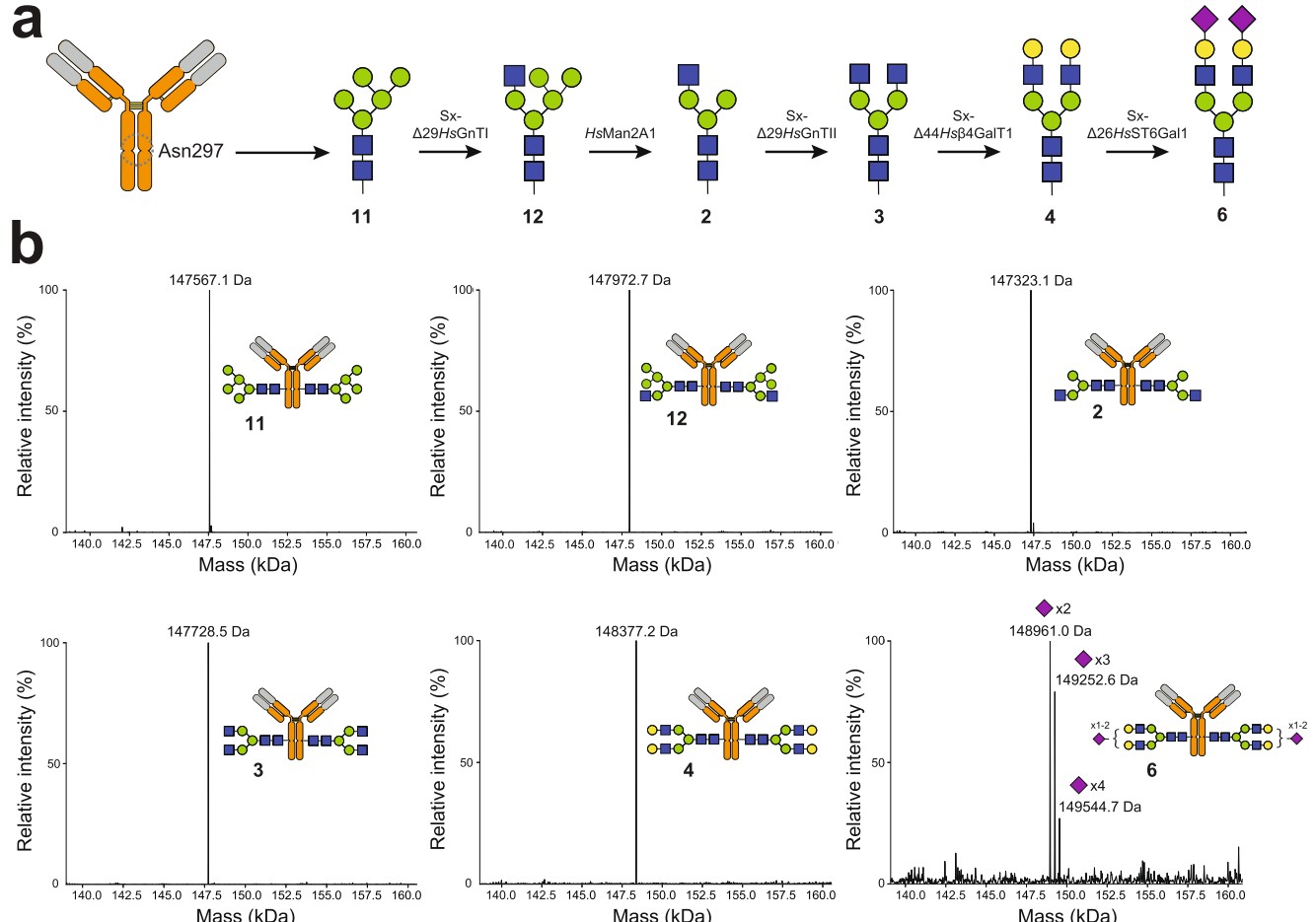

**Fig. 5 | Remodeling of IgG-Fc N-glycans on trastuzumab using Sx-GTs.**
**a** Schematic of bioenzymatic routes to hybrid- and complex-type N-glycan structures linked to asparagine 297 (N297) of the trastuzumab antibody. Trastuzumab bearing Man5GlcNAc2 glycan (M5; glycan 11) derived from glycoengineered HEK293F lacking GnTI activity was used as a glycan primer. Subsequent cell-free glycan remodeling reactions yielded the following N-glycan structures: 12 (M5+GlcNAc); 2 (G0-GlcNAc); 3 (G0); 4 (G2); and 6 (G2S2). Glycan notation follows

IgG glycan short naming system. For a complete glycan list with chemical structures, see Supplementary Table 2. SIMPLEx-reformatted GTs and glycosidase for each synthesis step are provided above reaction arrow. **b** Deconvoluted LC-MS spectra in 140–160 kDa range using intact antibody analysis of trastuzumab bearing glycan 11 as starting material and enzymatically derived product glycans 2–4, 6, and 12. Structures of anticipated N-glycan products are provided in each spectrum. Full MS spectra (0–200 kDa) for all structures are provided in Supplementary Fig. 13.

GlcNAcMan5GlcNAc2 glycan (glycan 12) directly on trastuzumab (Fig. 5b). The two terminal Man residues on the α1,6-man branch of 12 were then removed using human Golgi Man2A1 (HsMan2A1), yielding trastuzumab bearing glycan 2. Subsequent cell-free glycan remodeling reactions using Sx-Δ29HsGnTII and Sx-Δ44Hsβ4GalT1 furnished trastuzumab with glycans 3 and 4, respectively. Finally, Sx-Δ26HsST6Gal1 was used to cap glycan 4 with Neu5Ac, efficiently generating glycans 5 and, to a lesser extent, glycan 6 (Fig. 5b). Additional N-glycan structures including paucimannose (glycan 1), hybrid (glycan 13, 14), and complex (glycan 7, 15) types were also prepared directly on trastuzumab IgG-Fc using a variety of Sx-GTs (Supplementary Fig. 14). In most cases, glycan remodeling efficiency was near 100% following incubation with Sx-GTs for 16–36 h at 37 °C with ~80–90% recovery yield from purification between each step. Only the conversion to the di-sialylated N-glycan using Sx-Δ26HsST6Gal1 resulted in notably lower efficiency. While the reason for this inefficiency is unclear, it likely results from human ST6Gal1's known preference for the α1–3Man branch[52].

In addition to producing authentic, homogeneous human N-glycans, we also investigated whether Sx-GTs could generate IgG-Fc bearing unnatural glycan structures. To this end, we used Sx-Δ29HsGnTI to elaborate trastuzumab N-glycans with N-azidoacetylglucosamine (GlcNAz), a synthetic monosaccharide containing an

azide moiety (Supplementary Fig. 15, glycan 13) that served as a versatile chemical handle for regiospecific conjugation via bioorthogonal click chemistry. Indeed, it was possible to site-specifically modify this handle on trastuzumab with either a biotin group (glycan 17) or a fluorescent reporter (glycan 18), thereby providing a convenient route for extending the functional utility of Fc domain-linked glycans. Collectively, these results highlight the biocatalytic potential of SIMPLEx glycoenzymes in the construction of homogeneous glycans as both free and protein-linked structures and effectively pave the way for accelerating protein glycosylation studies as well as tailoring the biological, biophysical, and biomedical properties of glycoproteins.

## Discussion

In this work, we created a universal expression platform for producing nearly 100 different GTs, predominantly of human origin, at relatively high titers (~5–10 mg/L) using standard bacterial culture. The platform leverages SIMPLEx to engineer GT chimeras that are rendered highly soluble in the cytoplasm of E. coli cells. Consistent with earlier works[32,33], SIMPLEx-reformatted GTs retained biological activity as exemplified by the human ST6Gal1 chimera that exhibited activity that was similar to a commercially sourced enzyme. The ability to solubilize such a large set of GTs without compromising function made it possible to remodel the structures of different free and protein-linked

glycans including those found on the mAb trastuzumab. Overall, the platform described here represents a versatile addition to the synthetic glycobiology toolkit, providing easy access to a vast collection of transformative reagents that are expected to find use in structure-function studies of GTs and to fuel myriad applications where complex glycomolecules are featured.

Our previous studies revealed the capacity of SIMPLEx to broadly transform all major classes of IMPs into water-soluble molecules[32,33]. These IMPs included proteins having both bitopic and polytopic α-helical structures such as glutamate receptor (GluA2) and bacteriorhodopsin (bR) as well as polytopic β-barrel structures such as voltage-dependent anion channel 1. Here, this solubilization capacity was broadened to include polytopic α-helical GTs with multiple TMDs such as found in human mannosyltransferases Alg2, Alg3, and Alg12 and human glucosyltransferases Alg6, Alg8, and Alg10 as well as monotopic α-helical GTs with single-pass internal TMDs that could not be easily removed such as Alg2 and PigA. For these complex IMPs, introduction of an N-terminal decoy protein, MBP, prevented co-translational insertion of the polypeptide into the inner membrane through the signal recognition particle pathway[57] while the amphipathic ApoAI* domain effectively shielded the hydrophobic TMDs from the aqueous environment.

It is noteworthy that most of the GTs investigated here (72 out of 98 total) were simpler type II transmembrane proteins. Type II GTs such as *Hs*ST6Gal1 possess just a single-pass TMD at their N- or C-termini (Fig. 1a), which is generally not required for activity and is thus commonly removed during expression campaigns[18]. Hence, while the rationale for using SIMPLEx with full-length GTs including their TMDs was clear, it was less obvious that this solubilization method would benefit type II GTs lacking a TMD altogether. That said, removal of GTs from their transmembrane contexts and the lack of native interacting/folding partners can create difficulties in folding upon expression in heterologous hosts. Indeed, we observed that many N-terminally truncated GTs accumulated exclusively in the insoluble or detergent-solubilized fractions of *E. coli* cells, in agreement with numerous previous reports involving the expression of truncated GTs in bacteria. One possible reason for this poor expression is that GTs contain several moderately hydrophobic segments including around the stem region, just after the TMD, that can trigger unwanted membrane targeting or otherwise drive misfolding and aggregation. To circumvent this issue, fusion of solubility-enhancing partners such as MBP to the N-termini of truncated GTs is often obligatory, even in mammalian cells where GTs fused with GFP expressed significantly better than unfused versions of the same glycoenzymes[18]. However, introduction of fusion tags does not always lead to immediate success in terms of soluble GT expression and thus often requires lengthy optimization of growth and induction conditions as well as trial-and-error evaluation of different host strains and accessory factor (e.g., molecular chaperone) co-expression strategies[27,38,58]. Even when appreciable solubility is achieved, it is quite common for the resulting MBP fusions to accumulate as soluble but heterogeneous multimeric aggregates in which only a small fraction of the fusion protein is properly folded and active[27,38,59]. Along these lines, we observed >50% less activity for MBP-tagged Δ26*Hs*ST6Gal1 (lacking the native TMD) compared to its SIMPLEx counterpart, underscoring the essential contribution made by the ApoAI* domain in promoting solubility of a type II GT in the *E. coli* cytoplasm. We hypothesize that the amphipathic nature of ApoAI*, when expressed in the proximity of exposed hydrophobic patches in type II GT proteins, provides a stabilizing effect via hydrophobic interaction, akin to how ApoAI-based nanodiscs solubilize membrane proteins in solution[60].

Importantly, the SIMPLEx architecture enabled soluble expression for nearly 100 GTs (>95% "hit" rate) under standard, identically matched conditions without any optimization, thereby offering a universal solution to GT production in *E. coli* that has not been possible with stand-alone fusion tags such as MBP or other expression optimization techniques[61]. An additional layer of universality stems from the compatibility of SIMPLEx-mediated GT solubilization with other commonly used expression hosts such as yeast and HEK293 cells as well as with *E. coli*-based CFPS. Such platform flexibility is significant for several reasons. For one, each of these platforms is amenable to high-throughput profiling of protein expression and production that can be scaled up to larger volumes[62,63]. Moreover, in the case of yeast and HEK293, the compatibility of SIMPLEx-reformatted GTs in these well-established eukaryotic hosts may provide access to protein folding networks and PTMs including *N*- and *O*-linked glycosylation that may be important for the biological function of a subset of GTs[64] but are natively lacking in standard *E. coli* strains. In the case of *E. coli*-based CFPS, the "open" nature and multiplexability of these systems, combined with their speed and simplicity, should provide opportunities for high-throughput screening of GT function[65] as well as rapid discovery, prototyping, and optimization of glycomolecule synthesis pathways[66,67].

As proof of concept for the utility of our SIMPLEx pipeline, some of the solubilized products were used in coordinated cell-free reaction networks to catalyze the formation of chemically defined *N*-glycans. In one instance, it was possible to transform quantitative amounts of a simple paucimannose precursor *N*-glycan, Man$_3$GlcNAc$_2$ derived from glycoengineered *E. coli*[51,68], into complex biantennary *N*-glycans including those containing core-fucose and sialic acid caps using a set of SIMPLEx-reformatted GTs. This workflow to efficiently generate a library of complex *N*-glycans, starting from expression and purification and then finally utilization of SIMPLEx-reformatted GTs, could be completed in less than one week. Using an identical strategy, it was possible to generate a spectrum of homogenous *N*-glycan structures on intact glycoproteins including trastuzumab, a mAb therapy used to treat breast and stomach cancers. Akin to our earlier engineering of an artificial cytoplasmic disulfide formation pathway involving a water-soluble SIMPLEx variant of DsbB[33], ensembles of SIMPLEx-reformatted GTs could similarly be assembled into designer pathways, either in vitro or in living cells, for the on-demand biosynthesis of important glycans and glycoconjugates. Looking forward, we anticipate that the constructs, expression systems, and workflows for glycoenzyme production described here will find widespread use by those seeking to push the boundaries of our knowledge of glycobiology and glycochemistry and its application in health, energy, and materials science.

## Methods

### Strains and cell lines

All bacterial, yeast, and mammalian cells used in this study are listed in Supplementary Table 1. *E. coli* strain DH5α was used for all molecular cloning and plasmid storage. *E. coli* strain BL21(DE3) and its derivative SHuffle T7 Express *lysY* (New England Biolabs) were used for all protein expression and purification. LB medium was used to culture *E. coli* in all experiments and was supplemented with appropriate antibiotics for plasmid maintenance. The final concentration for each antibiotic used was: 50 µg/mL kanamycin, 20 µg/mL chloramphenicol, and 100 µg/mL ampicillin. Yeast strain SBY49 was kindly provided by Dr Scott Emr (Cornell University). Yeast cells were grown in complex yeast extract peptone dextrose medium or yeast nitrogen base medium without amino acids supplemented with uracil dropout amino acids (-URA media) for plasmid maintenance. HEK293T cells were obtained from ATCC (CRL-3216) and cultured in DMEM supplemented with 10% FetalClone (VWR), 4.5 g/L glucose and L-glutamine, and 1% (w/v) penicillin-streptomycin-amphotericin B (Thermo Fisher Scientific). FreeStyle™ 293-F cells (HEK293F) were obtained from Thermo Fisher Scientific (Cat # R79007). Expi293F™ GnTI⁻ cells (HEK293F GnTI⁻) were

obtained from Thermo Fisher Scientific (Cat # A39240) and were cultured in Expi293™ Expression Medium supplemented with 1% (w/v) penicillin-streptomycin-amphotericin B (Thermo Fisher Scientific). All cells were maintained in a 37 °C incubator with 5% $CO_2$ and 90% relative humidity. Authentication of each cell line used in this study included morphology analysis, PCR assays with species-specific primers, and STR profiling, the latter of which was performed using ATCC's human cell STR profiling service.

## Cell growth analysis

To facilitate high-throughput cell growth measurements, three individual colonies corresponding to each construct were seeded into 96-deep well plates (Eppendorf) where each well contained 100 µL LB media. Culture plates were then sealed using plate sealer and placed in an incubator shaker at 37 °C for 16 h. Then, 5 µL of the overnight culture was subcultured into fresh 100 µL LB media and incubated for 8 h, after which IPTG was supplemented to a final concentration of 0.1 mM. Protein expression proceeded at 16 °C for 18 h. To measure $OD_{600}$, 10 µL of each sample was mixed with 90 µL DI water in a Costar 96-well assay plate (Corning) and $OD_{600}$ of all samples was measured in an Infinite M1000Pro spectrophotometer (Tecan).

## Plasmid construction

All plasmids used in this study are listed in Supplementary Table 1. The collection of prokaryotic and eukaryotic glycoenzymes was selected from the CAZy database[19]. Amino acid sequences were all extracted from the UniProt database[69]. Each GT coding region was examined for membrane domains using the UniProt database to determine the TMD topology. GTs with internal or multi-pass TMDs were expressed as full-length proteins. For type II transmembrane proteins, N- and C-terminal TMD segments were truncated while stem regions were generally retained. For other classes, N-terminal signal peptides and C-terminal ER retention signals were generally removed. Amino acid sequences of full-length and truncated variants of all GTs in this study are provided in Supplementary Dataset 1. All GT genes were codon-optimized for expression in *E. coli* using GeneArt software (Thermo Fisher Scientific). These genes were then synthesized and ligated into the previously described SIMPLEx plasmid[32] to generate plasmids encoding SIMPLEx-reformatted GTs having the form pET28a(+)-MBP-(*Nde*I)-GT-(*Eco*RI)-ApoAI*−6xHis. PCR was used to amplify each GT gene with flanking *Nco*I and *Not*I restriction sites, and then ligated into pET28a(+) vector to create plasmids for expression of unfused GT constructs having the form pET28a(+)-(*Nco*I)-GT-(*Not*I)−6xHis. All PCR reactions were performed using 0.1 µM gene-specific primers, 50 ng DNA template, and Phusion® High-Fidelity DNA Polymerase (New England Biolabs). Ligation products were used to chemically transform *E. coli* DH5α, and the transformation cultures were plated on LB-agar plates containing kanamycin. Clones were selected and screened by colony PCR using 2x-OneTaq Quickload master mix (New England Biolabs). Successful clones were confirmed by Sanger sequencing at the Cornell Biotechnology Resource Center. Due to incompatibility of DNA restriction sites, plasmids used for expression in yeast and mammalian cells were constructed using Gibson assembly. Briefly, standard PCR was used to amplify target genes containing 20−25 bp homologous regions with vectors at both ends. 50 ng of linearized vector and 150 ng of amplified insert were then combined in a Gibson Assembly Master Mix (New England Biolabs) and incubated for 1 h. Assembly reactions were then used to transform *E. coli* DH5α, after which clones were screened and confirmed according to a similar procedure as described above.

## Small-scale expression and subcellular fractionation

Plasmids encoding Sx-GT and unfused GT constructs were used to transform either *E. coli* strain BL21(DE3) for GTs containing no disulfide bonds or SHuffle T7 Express *lysY* for GTs containing predicted or confirmed to contain disulfide bonds. Small 5-mL LB cultures of *E. coli*

harboring either an Sx-GT or GT plasmid were grown to an optical density at 600 nm ($OD_{600}$) of ~0.6−0.8 and then induced with IPTG to a final concentration of 0.1 mM. Protein expression proceeded for 18 h at 16 °C, after which culture volumes equivalent to $OD_{600}$ of 2.0 were harvested. Media was removed by centrifugation and the resulting cell pellet was resuspended in 1 mL phosphate buffer saline (PBS). Cells were lysed using a Q125 Sonicator (Qsonica) with a 3.175-mm diameter probe at a frequency of 20 kHz and 40% amplitude. Lysate was first centrifuged at 15,000 × g for 30 min at 4 °C. Supernatant was collected and centrifuged at 100,000 × g for 1 h at 4 °C. The supernatant from this ultracentrifugation step was collected as the soluble fraction. Pellet was then resuspended in 1 mL PBS containing 1% (v/v) Triton X-100. The suspension was incubated for 1 h at 4 °C to allow partitioning of membrane proteins into Triton X-containing buffer. Following ultracentrifugation at 100,000 × g for 1 h at 4 °C, supernatant was collected as the detergent-solubilized fraction, while the pellet was taken as the insoluble fraction.

## Protein purification and yield determination

A single colony of *E. coli* harboring plasmid DNA encoding a specific glycoenyzme was selected from a transformation plate and grown overnight in LB media at 37 °C. The next day, cells were subcultured 5% into 1 L of fresh LB media. Cells were grown at 37 °C until $OD_{600}$ reached ~0.6−0.8, after which IPTG was supplemented into the culture at 0.1 mM final concentration. Protein expression proceeded at 16 °C for 18 h. Unless otherwise noted, all purification procedures were performed at 4 °C. Cells were harvested, resuspended in PBS supplemented with 10% (v/v) glycerol, and lysed by passing the cell suspension through an Emulsiflex C5 homogenizer (Avestin) twice at 15,000 psi maximum pressure. Supernatant was collected following centrifugation at 15,000 × g for 30 min and then incubated with 300 µL pre-washed HisPur™ Ni-NTA resin (Thermo Fisher Scientific) at 4 °C for 1 h. The suspension was loaded onto an Econo-Pac® gravity flow chromatography column (Bio-Rad) and resin was washed with 6 column volumes HisPur wash buffer (50 mM $NaH_2PO_4$, 300 mM NaCl, 10 mM imidazole, pH 8.0). The target protein was eluted with HisPur elusion buffer (50 mM $NaH_2PO_4$, 300 mM NaCl, 300 mM imidazole, pH 8.0). Sample was then buffer exchanged into PBS using Zeba spin desalting columns, 7 K MWCO (Thermo Fisher Scientific). Protein concentration was determined using Bradford assay (Bio-Rad). Purified protein fractions were subjected to standard Coomassie-blue staining of SDS-PAGE gels and purity of each was determined by densitometry analysis using Bio-Rad Image Lab software (version 6.1.0 build 7), whereby the intensity of the band corresponding to the full-length Sx-GT construct was normalized to the intensity of all bands that appeared in the same lane of the gel. In general, purity of isolated Sx-GTs was ~50−80% following just a single-step Ni-NTA purification. Final yield values were tabulated based on both total protein concentration and purity, and were representative of three biological replicates starting from freshly transformed cells.

All other purification was performed as described above but with amylose resin (NEB) instead of Ni-NTA resin. Clarified lysate was incubated with 300 µL pre-washed amylose resin with rotation for 2 h at 4 °C. The suspension was loaded onto an Econo-Pac® gravity column (Bio-Rad) and resin was washed with 6 column volumes of amylose column buffer (20 mM Tris-HCl, 200 mM NaCl, 1 mM EDTA, pH 7.4). The target protein was eluted with amylose elusion buffer (10 mM maltose in column buffer). Protein purity and concentration were determined by Coomassie staining and Bradford assay (both from Bio-Rad), respectively. Proteins were kept at 4 °C for 2 weeks. For longer-term storage at −80 °C, protein solution was supplemented with 10% (v/v) glycerol and 0.02% (w/v) sodium azide as a cryogenic agent and bacteriostat, respectively.

For human MAN2A1 expression and purification, an expression construct encoding the truncated catalytic domain of human MAN2A1

(UniProt Q16706, residues 27–1144) was used[18]. This recombinant human MAN2A1 construct was expressed by transient transfection of suspension culture HEK293F cells, with soluble recombinant human MAN2A1 expressed as a soluble secreted product that was purified as described[70]. Briefly, the conditioned culture medium was loaded on a Ni$^{2+}$-NTA Superflow column (Qiagen) equilibrated with 20 mM HEPES, 300 mM NaCl, 20 mM imidazole, pH 7.4, washed with column buffer, and eluted successively with column buffers containing stepwise increasing imidazole concentrations (40–300 mM). The eluted fusion protein was pooled, concentrated, and concurrently mixed with recombinant TEV protease and EndoF1 at ratios of 1:10 relative to the GFP-MAN2A1 for each enzyme, respectively, and incubated at 4 °C for 36 h to cleave the tag and glycans. Dilution to lower the imidazole concentration was followed by passing the sample through a Ni$^{2+}$-NTA column to remove the fusion tag and His-tagged TEV protease and EndoF1. The protein was further purified on a Superdex 75 gel filtration column (GE Healthcare) and peak fractions of MAN2A1 were collected. The protein buffer was exchanged by ultrafiltration and adjusted to 1 mg/mL with buffer containing 20 mM HEPES, 100 mM NaCl, pH 7.0, 0.05% sodium azide, and 10% glycerol and stored at −80 °C until use.

For antibody expression and purification, glycoengineered HEK293F GnTI⁻ cells were used as follows. After at least three passages, cells were washed and resuspended at 3 million cells per mL concentration. Plasmid pVITRO1-Trastuzumab-IgG1/κ (Addgene #61883) was prepared from *E. coli* culture and the purified plasmid was flowed through an endotoxin removal column to remove contaminating endotoxin. Plasmid DNA-cationic lipid complex was then generated using Lipofectamine™ Transfection Reagent (Thermo Fisher Scientific) and was slowly added into the culture media with gentle mixing. The amount of DNA, cationic-lipid reagents, and cells were scaled linearly according to the manufacturer's protocol. Cells were maintained in a 37 °C incubator shaker for 24 h prior to being supplemented with Expression Enhancer Reagents (Thermo Fisher Scientific). Cell cultures were maintained at the same condition for another 5 days to allow antibody accumulation in the culture supernatant. Cells were then removed by centrifugation at 1000 × g for 5 min and supernatant was filtered through a 0.2-micron bottle-top filter. Supernatant was then mixed with 1× PBS at a 1:1 (v/v) ratio. This solution was flowed through MabSelect SuRe resin (Sigma-Aldrich) twice to allow antibody capture on protein A/G beads. Following extensive washing with 1× PBS, captured antibodies were eluted using glycine solution (pH 2.0) directly into neutralizing buffer (Tris-HCl pH 8.5). The antibody product was then buffer exchanged into 1× PBS supplemented with 0.01% sodium azide. Antibody was stored at 4 °C and was stable at the described conditions for at least a month.

## Immunoblot analysis

Prior to electrophoretic separation, samples were combined with NuPAGE™ 4X LDS Sample Buffer (Invitrogen) supplemented with 2.5% β-mercaptoethanol and then boiled at 100 °C for 10 min. Samples equivalent to $OD_{600}$ of 0.375 for small-scale expression or 15 μL of CFPS reaction were loaded into each well of Bolt™ 8% Bis-Tris Plus Gels (Thermo Fisher Scientific). Following electrophoretic separation and transfer to Immobilon-P polyvinylidene difluoride membranes (0.45 μm), blots were washed with TBS buffer (80 g/L NaCl, 20 g/L KCl, and 30 g/L Tris-base) followed by a 1-h incubation in blocking solution (50 g/L non-fat milk in TBS supplemented with 0.05% (v/v%) Tween-20; TBST). Blots were then washed four times with TBST in 10-min intervals and probed with primary antibodies including rabbit polyclonal antibody to 6xHis epitope tag (Thermo Fisher Scientific; Cat # PA1-983B; 1:5000 dilution), mouse monoclonal anti-GAPDH clone 6C5 (Calbiochem; Cat # CB1001; 1:10,000 dilution), rabbit polyclonal anti-GroEL (Sigma-Aldrich; Cat # G6532; 1:20,000 dilution), and rabbit anti-alpha tubulin clone EPR13799 (Abcam; Cat # ab184970; 1:10,000 dilution). Secondary antibodies were used as needed and these include goat anti-

rabbit IgG H&L (HRP) (Abcam; Cat # ab6721; 1:5000 dilution), rabbit anti-mouse IgG H&L (HRP) (Abcam; Cat # ab6728; 1:5000 dilution), and ExtrAvidin®–Peroxidase (Sigma-Aldrich; Cat # E2886; 1:4000 dilution). Blots were then washed as above. Imaging of blots was performed using a ChemiDoc™ XRS⁺ System following a brief incubation with Western ECL substrate (Bio-Rad).

## Sialyltransferase activity assay

Kinetic analysis of sialyltransferases was performed using a commercial sialyltransferase activity kit (R&D Systems, Cat # EA002) according to the manufacturer's protocols. Briefly, assays used 2 μg/mL of purified Sx-Δ26*Hs*ST6Gal1 or commercial human ST6Gal1 (amino acids 44–406) (R&D Systems; Cat # 7620-GT-010), 1.0 mg/mL of asialofetuin (Sigma-Aldrich; Cat # A4781-50MG) as acceptor substrate, and 0.02–0.8 mM of CMP-Neu5Ac as donor substrate. All reactions were incubated for 15 min at 37 °C. Values for $V_{max}$ and $K_m$ were determined using Prism 9 for MacOS version 9.2.0. A conversion factor used for calculating the amount of enzymatically released inorganic phosphate from CMP-Neu5Ac was determined to be 3833.5 pmol/$OD_{620}$ using the phosphate standards included in the kit and was used for all data analysis. Specific activity was calculated using 0.1 mM of CMP-Neu5Ac, 1.0 mg/mL of asialofetuin, and 0.04–0.23 μg of Sx-Δ26*Hs*ST6Gal1. A linear plot of absorbance ($OD_{620}$) versus amount of Sx-Δ26*Hs*ST6Gal1 was generated (Supplementary Fig. 2b). The slope of this plot was transformed using the conversion factor and divided by the reaction time to calculate the specific activity in units of pmol/min/μg.

## Bioorthogonal click chemistry-based chemoenzymatic remodeling

Strain-promoted alkyne-azide cycloaddition was used to assess the ability of Sx-GTs to chemoenzymatically remodel glycoprotein substrates. In a typical reaction, a 1.5-mL microcentrifuge tube was charged with 20 μL of reaction mixture consisting of 1 μg purified Sx-GT or 50 μg cell lysate, 3 μg purified acceptor glycoprotein substrate, and 10 mM nucleotide-activated monosaccharide donor modified with an azide functional group. Depending on the GT reactions, the nucleotide-activated monosaccharide donors included UDP-GlcNAz, UDP-GalNAz, GDP-AzFuc, and CMP-AzNeu5Ac (all from R&D Systems). Following an incubation in a 37 °C water bath for 1 h, reaction mixtures were supplemented with 2-iodoacetamide (Sigma-Aldrich) at 100 mM final concentration and incubated in the dark at room temperature for 1 h. Then, 100 mM final concentration of carboxyrhodamine 110 or biotin(PEG)$_4$ conjugated dibenzocyclooctyne-amines (Click Chemistry Tools) in *N,N*-dimethylformamide was supplemented into the reaction mixture. Strain-promoted click reactions were carried out at 37 °C for 2 h. Samples were then combined with 4X LDS Sample Buffer (Invitrogen) supplemented with 2.5% β-mercaptoethanol and heated at 65 °C for 5 min. Following SDS-PAGE analysis, in-gel fluorescence from carboxyrhodamine 110-linked glycans on glycoproteins was measured using a ChemiDoc™ MP Imaging System (Bio-Rad) with 501/523 nm $\lambda_{ex}$/$\lambda_{em}$. Biotin-linked glycans on glycoproteins were analyzed following immunoblot analysis using horseradish peroxidase-conjugated streptavidin (Sigma-Aldrich) in a similar manner as described above for immunoblot analysis.

## Cell-free protein synthesis

*E. coli* lysate was prepared according to an established protocol[71]. Briefly, *E. coli* strain BL21(DE3) was cultured in 2xYTPG media (16 g/L tryptone, 10 g/L yeast extract, 5 g/L NaCl, 7 g/L potassium phosphate monobasic, 3 g/L potassium phosphate dibasic and 18 g/L glucose) at 37 °C with 0.5 mM IPTG until $OD_{600}$ reached ~1.0. Cells were then harvested and washed twice with cold S30 buffer (10 mM tris-acetate pH 8.2, 14 mM magnesium acetate, and 60 mM potassium acetate). The resulting pellet was stored at −80 °C until used. To prepare crude extract, pellets were thawed on ice and resuspended with S30 buffer

(1 mL per gram cell pellet). Cells were lysed using a Q125 Sonicator with a 3.175-mm diameter probe at a frequency of 20 kHz and 40% amplitude until the total energy input reached 1500 J. Lysate was then centrifuged twice at 30,000 × g at 4 °C for 30 min. Supernatant was then collected, aliquoted, and stored at −80 °C until used. Cell-free synthesis of Sx-GT and unfused GT constructs was performed using the modified PANOx-SP system[72]. Specifically, S30 lysate was preconditioned with 750 μM iodoacetamide in the dark at room temperature for 30 min and then lysate was supplemented with 200 mM glutathione at a 3:1 ratio between oxidized and reduced forms. Then, 200 ng plasmid DNA was introduced into CFPS reaction containing 30% (v/v) S30 lysate and the following: 12 mM magnesium glutamate, 10 mM ammonium glutamate, 130 mM potassium glutamate, 1.2 mM adenosine triphosphate, 0.85 mM guanosine triphosphate, 0.85 mM uridine triphosphate, 0.85 mM cytidine triphosphate, 0.034 mg/mL folinic acid, 0.171 mg/mL *E. coli* tRNA (Roche), 2 mM each of 20 amino acids, 30 mM phosphoenolpyruvate (Roche), 0.33 mM nicotinamide adenine dinucleotide, 0.27 mM coenzyme-A, 4 mM oxalic acid, 1 mM putrescine, 1.5 mM spermidine, and 57 mM HEPES. The synthesis reaction was carried out at 30 °C for 6 h, after which the sample was centrifuged at 15,000 × g for 30 min at 4 °C. Supernatant was collected and stored at −20 °C until further analysis.

### Yeast and mammalian cell expression

Yeast cells were transformed with plasmid pYS338 encoding Δ26*Hs*ST6Gal1 using the LiAc/single-stranded carrier DNA/PEG method[73]. For yeast expression, SBY49 cells were grown in -URA media at 30 °C until $OD_{600}$ reached ~0.6–0.8, after which protein expression was induced with galactose to a final concentration of 2% (w/v). Protein expression was performed for 22 h at 30 °C. Yeast cells were lysed by vortexing the cell suspension with glass beads in PBS containing zymolyase enzyme. For mammalian cell expression, 2.0 mL of HEK293T cells at ~80% confluency in a 6-well plate were transfected with 2 μg plasmid DNA using jetPRIME® transfection reagent (Polyplus Transfection). After transfection, cells were maintained in an incubator at 37 °C with 5% $CO_2$ and 90% relative humidity for 36 h, after which they were harvested. HEK293T cells were lysed by tip sonication. Subcellular fractionation analysis for yeast and HEK293T cells was performed similarly as described above. All samples were stored at −20 °C until further analysis.

### Cell-free bioenzymatic glycan synthesis

All glycans and nucleotide-activated sugar substrate solutions were prepared in sterile DI water and stored at −20 °C. Glycan 1 was prepared as described[48]. Briefly, dried cell pellets from a 250-mL culture of *E. coli* Origami2(DE3) *gmd*::kan *ΔwaaL* cells carrying plasmid pConYCGmCB[68] were resuspended in 2:1 chloroform: methanol, sonicated, and the remaining solids collected by centrifugation. This pellet was sonicated in water and collected by centrifugation. The resulting pellet was sonicated in 10:10:3 chloroform: methanol:water to isolate the lipid-linked oligosaccharides (LLOs) from the inner membrane. The LLOs were purified using acetate-converted DEAE anion exchange chromatography as they bind to the anion exchange resin via the phosphates that link the lipid and glycan. The resulting compound was dried and treated by mild acid hydrolysis to release glycans from the lipids. The released glycans were then separated from the lipid by a 1:1 butanol:water extraction, wherein the water layer contains the glycans. The glycans were then further purified with a graphitized carbon column using a 0–50% water: acetonitrile gradient. Following this procedure, we reproducibly obtained ~750 μg of glycan 1 that was well resolved from contaminant peaks (Fig. 4b). To synthesize glycan 2, 5 μg of glycan 1 was incubated with 20 μg/mL Sx-Δ29*Hs*GnTI and 10 mM UDP-GlcNAc (Sigma-Aldrich) in GnT buffer (20 mM HEPES, 50 mM NaCl, 10 mM $MnCl_2$, pH 7.2) at 37 °C for 16 h. To synthesize glycan 3, glycan 2 was incubated with 80 μg/mL Sx-

Δ29*Hs*GnTII and 20 mM UDP-GlcNAc in GnT buffer at 37 °C for 36 h. Glycan 3 was then incubated with 20 μg/mL Sx-Δ44*Hs*β4GalT1 and 10 mM UDP-Gal (Sigma-Aldrich) in GalT buffer (20 mM HEPES, 150 mM NaCl, 10 mM $MnCl_2$, pH 7.5) at 37 °C for 16 h to produce glycan 4. Sialic acid terminal glycans 5 and 6 were synthesized by incubating glycan 4 with 20 μg/mL Sx-Δ26*Hs*ST6Gal1 and 20 mM CMP-Neu5Ac (Sigma-Aldrich) in SiaT buffer (50 mM sodium phosphate, 150 mM NaCl, 10 mM $MgCl_2$, pH 8.0) at 37 °C for 16 h. Glycan 7 was synthesized by incubating glycan 4 with 20 μg/mL Sx- Δ30*Hs*FucT8 and 10 mM GDP-fucose (Sigma-Aldrich) in FucT buffer (100 mM MES, 10 mM $MgCl_2$, pH 7.0) at 37 °C for 16 h. Glycans 8, 9, and 10 were synthesized sequentially from glycan 7 using Sx-Δ44*Hs*β4GalT1 and Sx-Δ26*Hs*ST6Gal1 as described above for glycans 4, 5, and 6. Following reaction clean-up and glycan purification, reaction progress was monitored by MALDI-TOF MS. Briefly, 1 μL (~25 ng) of partially purified glycan was co-crystalized with 1 μL matrix consisting of 2,5-dihydroxybenzoic acid (10 mg/mL) in 70% (v/v) acetonitrile. The sample was analyzed in positive mode MALDI-TOF (SCIEX TOF/TOF 5800) operated in linear mode with data acquisition at 2000 shots/spot in the 5–100-kDa mass range. Because sialic acid is subject to MS-induced in-source and metastable decay, successful biosynthesis of glycans 5, 6, 9, and 10 was verified by nano LC-MS/MS analysis as described below.

### Cell-free bioenzymatic glycan remodeling on glycoproteins

Unless noted otherwise, all glycoprotein remodeling reactions were performed at 37 °C for 1 h prior to bioorthogonal labeling reaction as described above. The sialyltransferase activity of Sx-*Cj*CstII was assessed using human A1AT as glycoprotein acceptor substrate. A total of 3 μg of recombinant A1AT (R&D Systems) was treated with 20 U/μL α2-3,6,8,9 neuraminidase A (NEB) in a 10-μL reaction at 37 °C for 2 h to remove terminal sialic acid residues on A1AT glycans. Reaction mixtures were then heated at 85 °C for 15 min to inactivate neuraminidase A. Neuraminidase A-treated A1AT was then incubated with Sx-*Cj*CstII and CMP-AzNec5Ac in SiaT buffer in a 37 °C water bath for 1 h. Sialyltransferase activity of Sx-Δ34*Hs*ST3Gal1 was evaluated in a similar manner but neuraminidase-treated bovine submaxillary glands mucin (Sigma-Aldrich) was used as the glycoprotein substrate. *N*-acetylglucosaminyltransferase activity of Sx-Δ29*Hs*GnTI was assessed using MBP-GCG[DQNAT], a fusion between *E. coli* MBP and human glucagon (residues 1–29) followed by a C-terminal DQNAT glycosylation tag[68]. The MBP-GCG[DQNAT] construct was glycosylated with $Man_3$-$GlcNAc_2$ using glycoengineered *E. coli* as described[68]. Briefly, Origami2(DE3) *gmd*::kan *ΔwaaL* cells carrying plasmid pConYCGmCB along with plasmid pMAF10[74] and pTrc-spDsbA-MBP-GCG[DQNAT][68] were grown in 100 mL of LB at 37 °C until $OD_{600}$ reached ≈1.5. Culture temperature was reduced to 30 °C and allowed to grow overnight at 30 °C. The next day, cells were induced with 0.1 mM IPTG to initiate synthesis of the MBP-GCG[DQNAT] acceptor protein. Protein expression proceeded for 8 h at 30 °C. Cells were then harvested and subjected to subcellular fractionation. This involved pelleting and washing 100 mL of IPTG-induced culture with subcellular fractionation buffer (0.2 M Tris-Ac (pH 8.2), 0.25 mM EDTA, 0.25 M sucrose, and 160 μg/mL lysozyme). Cells were resuspended in 1.5 mL subcellular fractionation buffer and then incubated for 5 min on ice and spun down. After the addition of 60 μL of 1 M $MgSO_4$, cells were incubated for 10 min on ice. Cells were spun down, and the supernatant was taken as the periplasmic fraction. To isolate glycoproteins, periplasmic fractions were subjected to affinity chromatography using HisPur™ Ni-NTA resin (Thermo Fisher Scientific). Eluates were collected, solubilized in Laemmli sample buffer containing 5% β-mercaptoethanol, and resolved on SDS-polyacrylamide gels. Purified MBP-GCG[DQNAT] was incubated with Sx-Δ29*Hs*GnTI and UDP-GlcNAz in GnT buffer in a 37 °C water bath for 1 h. Fucosyltransferase activity was evaluated by incubating A1AT or neuraminidase A-treated A1AT with Sx-Δ36*Hs*FucT7 and GDP-AzFuc in FucT buffer in a 37 °C water bath for 1 h.

## Endoglycosidase sensitivity assay

In a sterile Eppendorf microcentrifuge tube, 1 μg of purified trastuzumab bearing $Man_5GlcNAc_2$ glycan was incubated with: (i) *Streptococcus pyogenes* Endo S2 (Genovis # A0-GL8-020) in Glycobuffer 1 (NEB # B1727SVIAL); (ii) *Elizabethkingia meningosepticum* EndoF1 (Sigma-Aldrich #324725) in GlycoBuffer 4 (NEB #B1703); (iii) *Elizabethkingia miricola* Endo F3 (NEB #P0771S) in GlycoBuffer 4; or (iv) PBS control. Reaction mixtures were incubated at 37 °C for 16 h and the product was analyzed by LC-MS using intact protein MS mode.

## Cell-free bioenzymatic glycan remodeling on trastuzumab

Glycan remodeling on full-length mAb was performed in an on-column mode. In all, 50 μg purified trastuzumab bearing $Man_5GlcNAc_2$ glycan was first incubated with MabSelect SuRe resin (Sigma-Aldrich) for 10 min to allow antibody capture on protein A/G beads. This mixture was then transferred to a spin column, followed by washing twice with PBS. The bottom of the spin column was then capped with rubber cap. In a separate tube, 50 μL of a specific glycan remodeling reaction mixture was prepared. For preparing *N*-acetylglucosaminyltransferase, galactosyltransferase, fucosyltransferase, and sialyltransferase reaction mix, see above for details. UDP-GlcNAz substrate was used at the same concentration as UDP-GlcNAc. Reaction using β-*N*-acetylglucosaminidase S (NEB # P0744S) was performed in Glycobuffer 1 (NEB) at 37 °C for 4 h. Reactions using human Man2A1 mannosidase were performed in 50 mM sodium acetate buffer (pH 5.5) 1 mM $ZnCl_2$ at 37 °C for 16 h. Following each reaction step, the reaction mixture was removed by centrifugation at 300 × g for 2 min. Resin was then washed twice with PBS using the same centrifugation setting. In general, we observed ~80–90% recovery yield of IgG following purification as determined by NanoDrop spectrophotometer. Subsequent reaction mixture was then added to the column and the clean-up process was repeated for each reaction step. Final IgG product was eluted using glycine solution (pH 2.0) and analyzed immediately by LC-MS.

## Chromatography and mass spectrometry

Hydrophilic interaction liquid chromatography (HILIC) was carried out using an Exion HPLC system with built-in autosampler (SCIEX). The free glycan samples were reconstituted in buffer A (80%: 20% acetonitrile: water), filtered with a 0.22-μm spin filter (Corning), and loaded onto a Kinetex HILIC column (2.6 μm, 2.6 × 150 mm; Phenomenex) with 80% ACN/20% water as buffer A and 50 mM $NH_4FA$ with pH 4.4 as buffer B. LC was performed using a 7-min gradient from 80 to 0% of buffer B at a flow rate of 400 μL/min.

All LC-MS/MS analysis was carried out using an X500B QTOF (SCIEX) mass spectrometer equipped with an electrospray ion source and coupled with an Exion HPLC system. Each reconstituted sample was injected into a Kinetex HILIC column (2.6 μm, 2.6 × 150 mm; Phenomenex). The free glycans were eluted in a 9-min gradient of 80–0% (80% ACN/20% water) at 400 nL/min followed by a 3-min hold at 80% (80% ACN/20% water) for re-equilibration. The instrument was operated in positive ion mode with ESI voltage set at 5.0 kV, ion source gas 1, gas 2 = 50 psi, curtain gas = 35 and CAD gas = 7, and source temperature of 350 °C. Calibration was done using a positive calibrant with the CDS system. For free glycan analysis, the instrument was operated in MS full-scan mode from *m/z* range from 200 to 2000 followed by multiple reaction monitoring high-resolution (MRM-HR) scans from 0 to 12 min at two different collision energies of 20 and 35 V with DP = 20 V and accumulation time of 0.25 s. MS survey scans were performed for the mass range of *m/z* 200–2000 with DP = 20 V, CE = 7 V, and accumulation time of 0.25 s and MS/MS MRM-HR scans were at the same DP voltage and CE = 20 V, and with Q1 unit resolution. All MS and MS/MS raw spectra from each sample obtained by MRM-HR scan were analyzed by SCIEX OS 1.4 data analysis system. XIC spectra were extracted from MS full-scan with each MRM transition. The glycan structure was annotated manually using GlycanMass-ExPASy tool.

## Physicochemical data collection and analysis

The name, amino acid sequence, structure availability (full-length or partial), and predicted PTMs (i.e., disulfide bonds, glycosylation) for each GT enzyme were retrieved from the UniProt database[69]. GT family members were annotated from the CAZy database[19]. Amino acid sequences of full-length, truncated, and SIMPLEx-fused GTs were compiled in FASTA format. The $M_w$ and pI were calculated using the ExPASy Bioinformatics resource portal in average resolution setting[75]. Solubility prediction score was calculated using CamSol Intrinsic version 2.1[76]. The expression scores for all constructs were annotated based on immunoblots in Supplementary Fig. 7. Correlation between protein properties ($M_w$, pI, solubility prediction score, and expression score) was analyzed using R software version 3.4.2. Specifically, scatter plots between protein properties, generated with data points colored according to expression score, were used to examine any possible correlations. For the correlation between expression score and pI, datasets were analyzed as a function of expression score movement. Scatter plots comparing the pI of Sx-GT versus GT constructs were created and the data were colored according to the change in expression score. A similar approach was used to analyze the correlation between expression score and solubility prediction score. Because no statistical significance was observed for the correlation between expression score and either pI or solubility prediction score, general observations from the plots were described instead. For the correlation between expression score and $M_w$, data were categorized into three groups: $M_w < 40$ kDa, $M_w = 40$–$60$ kDa, and $M_w > 60$ kDa, before average expression score for each group was calculated. Both Sx-GT and GT constructs were binned using the same criteria since the added mass from the SIMPLEx fusion was constant for all constructs. Welch's *t*-test was used to analyze the statistical significance of categorical datasets.

## Statistics and reproducibility

To ensure the robust reproducibility of all results, experiments were performed with at least three biological replicates and at least three technical measurements. Sample sizes were not predetermined based on statistical methods but were chosen according to the standards of the field (at least three independent biological replicates for each condition), which gave sufficient statistics for the effect sizes of interest. All data were reported as average values with error bars representing the standard error of the mean. Statistical significance was determined by Welch's *t*-test and *p* values of <0.05 were considered significant. All graphs were generated using Microsoft Excel, Prism 9 for MacOS version 9.2.0, or R software version 3.4.2. No data were excluded from the analyses. The experiments were not randomized. The Investigators were not blinded to allocation during experiments and outcome assessment.

## Reporting summary

Further information on research design is available in the Nature Research Reporting Summary linked to this article.

# Data availability

All data generated or analyzed during this study are included in this article or the Supplementary Information/Source Data files provided with this paper. All unique materials used are readily available from the authors. Source Data are provided with this paper.

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

## Acknowledgements

We would like to thank Dr Bernard Henrissat for providing statistical data of the GT genes from the CAZy database. We thank Dr Scott Emr for providing the yeast strain and corresponding expression vector used in these studies. We thank Dr Sudeep Banjade, Dr May Taw, and Dr Morgan Ludwicki for their assistance and critical discussions regarding mammalian cell expression. We thank Dr Yimon Aye and Dr Weston Kightlinger for critical discussions of the manuscript and Dr Ashty Karim (ORCID# 0000-0002-5789-7715) for scientific communication consultation. This work was supported by the Bill and Melinda Gates Foundation (OPP1217652 to M.P.D. and M.C.J.), Defense Threat Reduction Agency (HDTRA1-15-10052 and HDTRA1-20-10004 to M.P.D. and M.C.J.), National Science Foundation (CBET-1159581, CBET-1264701, CBET-1936823 to M.P.D. and MCB-1413563 to M.P.D. and M.C.J.), and National Institutes of Health (1R01GM137314 and 1R01GM127578 to M.P.D. and R01GM130915 to K.W.M.). The work was also supported by seed project funding (to M.P.D.) through the National Institutes of Health-funded Cornell Center on the Physics of Cancer Metabolism (supporting grant 1U54CA210184). The content is solely the responsibility of the authors and does not necessarily represent the official views of the National Cancer Institute or the National Institutes of Health. T.J. was supported by a Royal Thai Government Fellowship and a Cornell Fleming Graduate Scholarship.

## Author contributions

T.J. designed research, performed all research, analyzed all data, and wrote the paper. Y.H.K., Y.L., O.Y., M.L., and D.G.C. designed research and performed research. R.B. and J.D.W. performed mass spectrometry analysis and aided in data interpretation. K.W.M., M.C.J., and D.M. designed the research and wrote the paper. M.P.D. directed research, analyzed data, and wrote the paper. All authors read and approved the final manuscript.

## Competing interests

M.P.D. has a financial interest in Gauntlet Bio, Inc., Glycobia, Inc., SwiftScale Biologics, Inc., Versatope, Inc., and UbiquiTx, Inc. M.C.J. has a financial interest in Gauntlet Bio, Inc. and SwiftScale Biologics, Inc.. M.P.D.'s and M.C.J.'s interests are reviewed and managed by Cornell University and Northwestern University, respectively, in accordance with their conflict-of-interest policies. M.P.D. and M.C.J. have no non-financial competing interests to declare. All other authors declare no competing interests.
