## [Peer Review File · Nature Communications]

REVIEWER COMMENTS

Reviewer #1 (Remarks to the Author):

Manuscript: NCOMMS-22-01743

Authors: Thapakorn Jaroentomeechai, Yong Hyun Kwon, Yiwen Liu, Olivia Young, Ruchika Bhawal, Joshua D. Wilson, Mingji Li, Digantkumar G. Chapla, Kelley W. Moremen, Michael C. Jewett, Dario Mizrahi, and Matthew P. DeLisa*

Title: A universal glycoenzyme biosynthesis pipeline that enables efficient cell-free remodeling of glycans

Comments for the authors:

The manuscript by DeLisa et al. tackles a major challenge for the application of mammalian glycosyltransferases (GTs): most of these glycosyltransferases cannot or hardly be produced as soluble proteins in bacterial expression hosts like *E. coli*. This also significantly limits the large-scale production of mammalian GTs. Even though microbial GTs serve as enzyme substitutes for the synthesis of glycans and glycoconjugates, they also face the problem of insufficient gene expression and protein folding resulting in the formation of inclusion bodies. Mammalian GTs are type II membrane proteins with one transmembrane domain (TMD) or proteins with multiple TMDs. These TMDs have often a function for subcellular localization of GTs, e.g. in the ER or Golgi apparatus.

The authors present here their previously published SIMPLEx protein engineering method as a universal strategy for the construction of fusion proteins of mammalian and also some microbial GTs. GT fusion proteins with an N-terminal decoy protein (e.g. MBP) and a C-terminal amphipathic protein (ApoAI*) are constructed to prevent membrane insertion and to shield hydrophobic surfaces from the aqueous environment. The authors demonstrate impressively the production of 95 soluble proteins out of 98 GTs. Selected, and formerly regarded as "difficult to express", GTs are characterized for their ability to transfer azido-functionalized donor nucleotide sugars. The peculiarity of the SIMPLEx strategy lies also in its applicability for GT expression in a eukaryotic host (e.g. yeast, HEK293T) to facilitate posttranslational modifications for functional GTs. Most importantly, the authors demonstrate the application of GTs in glycan remodeling of pharmaceutical important glycoproteins, e.g., trastuzumab). In a bioinformatic analysis, the authors classify the GTs according to their solubility scores calculated by the web-based tool Prot-Sol. They conclude that SIMPLEx facilitates the expression of high molecular weight GTs. Correlations of the solubility cores with pI or amino acid composition were not observed.

In conclusion, the results represent a breakthrough and great progress in the field of GT production for their application in glycoengineering. Further studies can be expected for optimizing and fine-tuning the production of active GTs.

The manuscript is well written and the presentation and discussion of the results are concise. There are the following points that need further clarification for subsequent conclusions:

1. GT activity assays:

– Page 7, lines 14-25: "... quantifying sialyltransferase activity in vitro."

– Page 26, lines 10 – 29: Bioorthogonal click chemistry-based GT assay

The described method gives hardly enzyme activity as defined by the rate: 1 unit= $\mu\text{Mol}/\text{min}$.

Most important for evaluation and comparison of active GTs (Sx-GTs and GTs) in the soluble E. coli lysates is the specific activity: U/mg protein. The authors should include these numbers and discuss their results based on U/mg protein. see comments in the supplementary part. Are there any "activity" numbers derived from the calibration curve? (Supp. Figure 2c; see comments)?

– Supp. Figure 2: Functional characterization of Sx- Δ 26HsST6Gal1.

What is the incubation time for the first step with CMP-Neu5Ac/CMP-azido-Neu5Ac? - also for Supp. Fig 2C?

Supp. Figure 2a: There is no measure that all Gal residues on the N-glycans are occupied by Neu5Ac/azido-Neu5Ac under assay conditions? The authors should demonstrate this by an N-glycan analysis (CE-LIF or HPLC with MS analysis).

Supp. Figure 2c: The standard curve for sialyltransferase activity should be depicted as a graph with regression analysis. Ideally the fluorescence signal should be linear to the enzyme concentration. This is not the case. How was the amount of active enzyme calculated: The author should give the formula for the calculation of enzyme units.

– Page 7, lines 26-29

Using clarified lysate generated from E. coli cells expressing Sx- Δ 26HsST6Gal1 as a catalyst source, we detected strong fluorescence from the labeled A1AT, indicating the presence of high sialyltransferase activity in the lysate (Fig. 1c; estimated to be 0.66 29 units per liter shake flask culture).

How are these units 0.66 U calculated from the fluorescence intensity? (see comment for Supp. Figure 2c)

2. Figure 1c:

To compare all enzyme constructs, their specific enzyme activities should be given (U/mg protein). See here also the comments for Supp. Figure 2.

3. Figure 2d: legend

Fluorescence intensity from a labeling experiment shall not be given as "enzyme activity". This is not what an enzymologist expects to see.

Here the ability to transfer azide-modified sugars from their nucleotide-activated donor substrates onto a glycoprotein is demonstrated – This is more a "functional characterization" unless activity numbers are given.

See here also the comments for Supp. Figure 2.

The authors should depict the specific enzyme activity (U/mg) for different glycoprotein concentrations.

The legend in Fig.1d is not clear: assign enzymes to the colored data points. What is rhd26HsST6GalT1? Is this the same control enzyme as used for the standard curve in Supp. Figure 2?

The X-axis depicts "substrate concentration"? Does "substrate" correspond to the tested glycoprotein substrate – which one?

Most of the data points (conc./AU) cannot be resolved between 0- 5 μ g/mL. Please zoom these data points.

4. Page 10, lines 19-21 and Figure 2

The number 95 is too optimistic when inspecting Figure 2

At least 16 Sx-GT samples show no or only a faint band in the upper panel of Figure 2.

The number should be corrected.

5. Page 15, lines 20-30 and Figure 4

The authors should indicate the expression hosts for the applied Sx-GTs. The FucT8 appears as a faint band in Figure 2f (FUT8), and this raises the question of whether the E.coli lysate contains an effective FucT8 activity.

The authors should assign (ii) Sx- Δ 29HsGnT1 and (iii) Sx- Δ 29HsGnT2 to those lysates shown in Figure 2d.

6. Page 17 & 18 and Figure 5

The authors should assign the Sx-GTs depicted in Figure 5a to those lysates shown in Figure 2d.

7. Page 25, lines 3-8:

What was the expression vector for yeast?

Protocol for the transformation of yeast cells is missing.

Reviewer #2 (Remarks to the Author):

Jaroentomeechia et al. describe how the SIMPLEx approach can be used to produce glycosyltransferases (GTs, incl. many human ones) in E. coli (as well as in yeast and HEK cells). The use of the SIMPLEx approach enables to produce rapidly and cost-effectively functional GTs. Although the SIMPLEx approach has been described before, the current study shows even more convincingly than the previous ones how extremely valuable it is. A subset of the produced Sx-GTs is subsequently successfully used for cell-free remodelling of (free and protein-linked) glycans. This study creates the foundation for a unique and versatile glycoengineering toolkit.

Minor questions/comments

-In the immuno-blotting no whole cell lysates are included. Why?

-Is it possible that the 'Sx fusion partners (MBP and ApoAI)' facilitate the transfer of GTs during immuno-blotting. Have the authors also used Coomassie stained gels to compare the production of proteins with and without the 'Sx fusion partners'?

-Long term storage conditions are described in the Materials and Methods section: is anything known about how long Sx-GTs can be stored without losing (too much) activity?

-Can the authors just like for the experiment shown in Figure 4 also comment on the efficiencies of the experiment shown in Figure 5?

-How pure were the isolated Sx-GTs? In the Materials and Methods section it is mentioned that the purity was assessed using SDS-PAGE followed by Coomassie staining. Can the authors show some representative examples in the supplement?

-In Figure 1d and SF6: what is the impact of the MBP and ApoAI moieties? If you have an equal amount of Sx-GT and GT protein, the GT protein should in principle contain more enzymatic activity? Should one somehow correct for the MBP and ApoAI moieties when comparing the activity/expression of Sx-GTs and GTs? Or does this referee totally miss something? In Figure 1d: what represents dark green and what represents light green?

Reviewer #3 (Remarks to the Author):

This manuscript entitled A universal glycoenzyme biosynthesis pipeline that enables efficient cell-free remodeling of glycans by Thapakorn Jaroentomeechai et al describes a novel glycosylation remodeling methods using E. coli cell-free system.

Pros:

- 1. It is a massive piece of work with novelty. The language is appropriate and easy to understand.**
- 2. This novel glycoengineering method indeed has the potential to flexibly and precisely tailor N-glycans on recombinant proteins in cell-free systems.**

Cons:

1. The first part about generating soluble Sx-HsST6Gal1 and Sx-Δ26HsST6Gal1 expressions with functional enzymatic activity verification is convincing and solid. But in Figure 1, miss annotations about the green and orange dots in Figure 1d.

2. Then moving forward, the second part for generating massive soluble expression of diverse GTs is impressive but I have questions. Without an enzymatic assay for each Sx-GT, it is hard to believe these engineered GTs originated from prokaryotic and eukaryotic organisms, are functional. Relevant glyco-enzyme activity assay like the one shown in Figure 1 or other methods is necessary. Moreover, any explanation on the low or no expression of some engineered GTs in Figure 2? Figure 2C has a big and continuous overexposed band, please explain. Furthermore, it is recommended to point out if the MW size of each engineered GT is right or not.

3. This work is with a focus on the cell-free system using E.coli lysate, but it is unclear the applicability of this pipeline in other platforms, as shown in Figure 3, Saccharomyces cerevisiae strain SBY49 and human embryonic kidney (HEK) 293T cells (I assume the authors used SBY49 and HEK cells instead of their lysate) are appropriate for this manuscript or not. It is not clear if SBY49 and HEK in this work are cells or lysates. If those are cell systems, then Figure 3 is somehow irrelevant to the topic of this work. Moreover, in Figure 3, only overexpression of Sx- Δ26HsST6Gal1 without testing multiple engineered GTs is too early to call that this pipeline is compatible with other systems.

4. Mass Spec methods:

Free glycans: Free glycan samples were analyzed by LC-MS/MS. Add time plot or LC plot to the MS plot in Figure 4 to see a complete picture. A full list of glycan identified is necessary as well as these glycan abundances resulting from each glyco-reaction. Moreover, some small peaks in Figure 4b were didn't annotated.

In the method section, it says "For free glycan analysis, the instrument was operated in MS full-scan mode 5 from m/z range from 2,00-2,000 followed by multiple reaction monitoring high-resolution 6 (MRM-HR) scan from 0-12 min at two different collision energies of 20 and 35 V with DP 7 = 20 V and accumulation time of 0.25 s.", while in Figure 4, bi-galactosylated and bi-sialylated N-glycans were identified at M/Z above 2000.

5. The significantly different band intensity of internal controls on WB blots: Supplemental Figure 1 and supplemental figure 3.

Response to Reviewer Comments

We thank the editor and reviewers for their thoughtful comments regarding our manuscript. In the text that follows, we have responded to each of these comments in a point-by-point fashion (reviewer comments are in black font while our responses are in blue italic font) and have made corresponding revisions to our manuscript. All revisions have been marked in red font in the revised manuscript. We believe these changes have significantly improved the manuscript such that it will appeal to the broad readership of Nature Communications.

Reviewer #1 (Remarks to the Author):

Comments for the authors:

The manuscript by DeLisa et al. tackles a major challenge for the application of mammalian glycosyltransferases (GTs): most of these glycosyltransferases cannot or hardly be produced as soluble proteins in bacterial expression hosts like *E. coli*. This also significantly limits the large-scale production of mammalian GTs. Even though microbial GTs serve as enzyme substitutes for the synthesis of glycans and glycoconjugates, they also face the problem of insufficient gene expression and protein folding resulting in the formation of inclusion bodies. Mammalian GTs are type II membrane proteins with one transmembrane domain (TMD) or proteins with multiple TMDs. These TMDs have often a function for subcellular localization of GTs, e.g. in the ER or Golgi apparatus.

The authors present here their previously published SIMPLEx protein engineering method as a universal strategy for the construction of fusion proteins of mammalian and also some microbial GTs. GT fusion proteins with an N-terminal decoy protein (e.g. MBP) and a C-terminal amphipathic protein (ApoA1*) are constructed to prevent membrane insertion and to shield hydrophobic surfaces from the aqueous environment. The authors demonstrate impressively the production of 95 soluble proteins out of 98 GTs. Selected, and formerly regarded as “difficult to express”, GTs are characterized for their ability to transfer azido-functionalized donor nucleotide sugars. The peculiarity of the SIMPLEx strategy lies also in its applicability for GT expression in a eukaryotic host (e.g. yeast, HEK293T) to facilitate posttranslational modifications for functional GTs. Most importantly, the authors demonstrate the application of GTs in glycan remodeling of pharmaceutical important glycoproteins, e.g., trastuzumab). In a bioinformatic analysis, the authors classify the GTs according to their solubility scores calculated by the web-based tool Prot-Sol. They conclude that SIMPLEx facilitates the expression of high molecular weight GTs. Correlations of the solubility scores with pI or amino acid composition were not observed.

In conclusion, the results represent a breakthrough and great progress in the field of GT production for their application in glycoengineering. Further studies can be expected for optimizing and fine-tuning the production of active GTs.

The manuscript is well written and the presentation and discussion of the results are concise. There are the following points that need further clarification for subsequent conclusions:

We appreciate Reviewer 1's positive assessment of our work and recognition that “the results represent a breakthrough and great progress in the field of GT production for their application in glycoengineering.” In the section that follows, we have addressed the aspects that Reviewer 1 encouraged us to consider.

1. GT activity assays:

- Page 7, lines 14-25: “... quantifying sialyltransferase activity in vitro.”
- Page 26, lines 10 – 29: Bioorthogonal click chemistry-based GT assay

1. The described method gives hardly enzyme activity as defined by the rate: 1 unit= $\mu\text{Mol}/\text{min}$. Most important for evaluation and comparison of active GTs (Sx-GTs and GTs) in the soluble E. coli lysates is the specific activity: U/mg protein. The authors should include these numbers and discuss their results based on U/mg protein. see comments in the supplementary part. Are there any "activity" numbers derived from the calibration curve? (Supp. Figure 2c; see comments)?

We agree completely with Reviewer 1 about the importance of providing more careful analysis of sialyltransferase activity. Because this was not possible with our bioorthogonal click chemistry-based assay without extensive assay validation, we instead performed a conventional sialyltransferase activity assay and added these data to the Results section of our revised manuscript. Specifically, the Sx- $\Delta 26\text{HsST6Gal1}$ enzyme was purified and subjected to kinetic analysis using a commercial kit (R&D Systems, catalog # EA002) for quantifying release of nucleotide cytidine 5'-monophosphate (CMP) from the donor substrate CMP-N-acetylneuraminic (CMP-Neu5Ac). The apparent K_M and V_{max} values for our solubilized Sx- $\Delta 26\text{HsST6Gal1}$ construct were $0.19 \pm 0.03 \text{ mM}$ and $85.5 \pm 6.3 \text{ pmol}/\text{min}$, respectively. These values were in close agreement with the apparent kinetic parameters that we measured for the commercial recombinant human ST6Gal1 (R&D Systems; catalog # 7620-GT-010) (see revised Figure 1c) and also in close agreement with measurements made previously by Boons and coworkers (Mbuja et al., 2013 Angew Chem Int Ed Engl). The specific activity of the Sx- $\Delta 26\text{HsST6Gal1}$ enzyme was $516.9 \text{ pmol}/\text{min}/\mu\text{g}$, which was consistent with previously published data for the unfused HsST6Gal1 enzyme (Houeix and Cairns, 2019 PeerJ).

- Supp. Figure 2: Functional characterization of Sx- $\Delta 26\text{HsST6Gal1}$. What is the incubation time for the first step with CMP-Neu5Ac/CMP-azido-Neu5Ac? - also for Supp. Fig 2C? Supp. Figure 2a: There is no measure that all Gal residues on the N-glycans are occupied by Neu5Ac/azido-Neu5Ac under assay conditions? The authors should demonstrate this by an N-glycan analysis (CE-LIF or HPLC with MS analysis). Supp. Figure 2c: The standard curve for sialyltransferase activity should be depicted as a graph with regression analysis. Ideally the fluorescence signal should be linear to the enzyme concentration. This is not the case. How was the amount of active enzyme calculated: The author should give the formula for the calculation of enzyme units.

The Methods section of our original manuscript included details about incubation time for the first step of the functional characterization assay, which was performed for 1 h at 37°C , after which reaction mixtures were supplemented with 100 mM 2-iodoacetamide and incubated in the dark at room temperature for 1 h. Then, 100 mM final concentration of carboxyrhodamine 110 or biotin(PEG)₄ conjugated dibenzocyclooctyne-amines in DMF was supplemented into the reaction mixture and strain-promoted click reactions were carried out at 37°C for 2 h. We have made minor revisions to the Methods section and to the schematic in Supplementary Figure 2c of our revised manuscript to more clearly indicate the timing of these steps.

Reviewer 1 correctly points out that we did not determine whether all Gal residues on the N-glycans are occupied by Neu5Ac/azido-Neu5Ac. However, these bioorthogonal click chemistry-based experiments have been re-prioritized simply as functional characterizations and not as enzymatic activity (see our response to comment 5 below). Hence, we do not feel that detailed characterization of the reaction products as proposed by Reviewer 1 is necessary and would not materially change the conclusions drawn about the practical utility of solubilized Sx-GTs for chemoenzymatic remodeling of protein-linked N-glycans. Furthermore, the extent of azido-Neu5Ac incorporation is not only a function of the sialyltransferase activity but also a function of Neu5Ac removal by neuraminidase A. Thus, it would be impossible to link any information about whether all Gal residues on the N-glycans are occupied by Neu5Ac/azido-Neu5Ac with the performance of our Sx-GT. Likewise, because we are no longer using the bioorthogonal click chemistry-based assay to quantify enzymatic activity, it is no longer necessary to extract information about specific enzyme activity (i.e., units/mg) from these experiments. Instead, these click experiments are now included for the purpose of showing the utility

of our engineered enzymes in carrying out chemoenzymatic remodeling of existing protein-linked N-glycans. For accurately measuring enzymatic activity, we have performed kinetic analysis using a standard commercial kit and provided this new data in Fig. 1c as discussed in comment 1 above.

- Page 7, lines 26-29. Using clarified lysate generated from E. coli cells expressing Sx- Δ 26HsST6Gal1 as a catalyst source, we detected strong fluorescence from the labeled A1AT, indicating the presence of high sialyltransferase activity in the lysate (Fig. 1c; estimated to be 0.66 29 units per liter shake flask culture). How are these units 0.66 U calculated from the fluorescence intensity? (see comment for Supp. Figure 2c)

As written above, we have now re-prioritized the bioorthogonal click chemistry-based experiments simply as functional characterizations to reveal the potential of Sx-GTs for chemoenzymatic remodeling of protein-linked N-glycans. Importantly, these click experiments are no longer being used to demonstrate enzymatic activity; hence, we have removed mention of units of activity from the manuscript. This is also appropriate because these results were generated with clarified lysate and not purified enzymes; therefore, mention of enzyme units in this context is inappropriate. Instead, we have added carefully performed kinetic analysis using purified enzymes to the manuscript (Fig. 1c), which allowed us to report apparent kinetic parameters and specific enzyme activity of our solubilized Sx- Δ 26HsST6Gal1 construct.

2. Figure 1c

To compare all enzyme constructs, their specific enzyme activities should be given (U/mg protein). See here also the comments for Supp. Figure 2.

Please see our comments above about specific enzyme activities in the context of the bioorthogonal click chemistry-based experiments. In the revised manuscript, we now present specific enzyme activity that was calculated from data generated using a commercial kit with purified enzymes, not clarified lysates.

3. Figure 2d: legend

Fluorescence intensity from a labeling experiment shall not be given as "enzyme activity". This is not what an enzymologist expects to see. Here the ability to transfer azide-modified sugars from their nucleotide-activate donor substrates onto a glycoprotein is demonstrated – This is more a "functional characterization" unless activity numbers are given. See here also the comments for Supp. Figure 2. The authors should depict the specific enzyme activity (U/mg) for different glycoprotein concentrations.

We agree with Reviewer 1 that the bioorthogonal click chemistry-based assay is more a "functional characterization" and should not be given as "enzyme activity". Indeed, to address this concern, we now present detailed kinetic analysis of our solubilized Sx- Δ 26HsST6Gal1 construct that was generated using a commercial kit with purified enzymes. This allowed us to report apparent kinetic parameters and specific enzyme activity (pmol/min/ μ g) of our solubilized Sx- Δ 26HsST6Gal1 construct. We have carefully revised the Results and Methods sections as well as relevant Figure Captions of our revised manuscript to reflect this re-prioritization of the bioorthogonal click chemistry-based assay. Specifically, we now refer to these experiments as "functional characterization" or "chemoenzymatic modification" instead of "enzyme activity", since this is more in line with what an enzymologist would expect to see.

The legend in Fig.1d is not clear: assign enzymes to the colored data points.

What is rhd26HsST6GalT1? Is this the same control enzyme as used for the standard curve in Supp. Figure 2?

We have fixed this labeling error in the revised manuscript (now Figure 1c), assigning the appropriate colors to the curves corresponding to each enzyme. The naming, rhd26HsST6Gal1, is for the commercial ST6Gal1 enzyme. We have changed this naming and revised the figure caption to make this clearer.

The X-axis depicts "substrate concentration"? Does "substrate" correspond to the tested glycoprotein substrate – which one?

Yes, the x-axis corresponds to the tested glycoprotein substrate, NA-treated A1AT. We have changed the x-axis label to indicate this more clearly.

Most of the data points (conc./AU) cannot be resolved between 0- 5 µg/mL. Please zoom these data points.

To better show the data points in the 0-5 µg/mL range, we have added an inset graph that depicts the data as a logarithmic plot.

4. Page 10, lines 19-21 and Figure 2.

The number 95 is too optimistic when inspecting Figure 2. At least 16 Sx-GT samples show no or only a faint band in the upper panel of Figure 2. The number should be corrected.

Reviewer 1 is correct that some of the counted Sx-GT samples show only a faint band. We arrived at 95 by counting all samples that gave a visible band, even if it was faint. The only samples that were not counted were the ones for which there was no visible band. It should be noted that we counted unfused GT samples exactly the same way, including those that showed only a faint band. We feel that reporting numbers this way is fair because we only tested one standard growth/induction condition to generate Figure 2 (starting $OD_{600} \approx 0.6$, induction with 0.1 mM β -D-1-thiogalactopyranoside (IPTG) at 16 °C for 16-20 h in LB medium) that was identical for each construct and did not include any optimization trials. It is important to note that optimization is commonly performed for recombinant glycosyltransferase expression campaigns employing bacteria (for example, Ortiz-Soto and Seibel, PLoS ONE 2016; Du et al, Cell Chem Biol 2019; Shu et al, Int J Mol Sci 2020). It is our opinion that with a small bit of expression optimization many/all of those faintly expressed constructs could be significantly improved. Indeed, a few of the enzymes used in Figures 4 and 5 were faintly expressed in lysates (e.g., FucT8), but could be improved by simple changes to the growth/induction conditions and also benefitted from the enrichment afforded by NiNTA purification. Thus, we have decided to keep the number at 95. However, to more accurately describe the complete results presented in Figure 2, we have revised the manuscript text as follows:

“Importantly, 95 of the Sx-GT constructs showed clearly visible accumulation in the soluble cytoplasmic fractions, with most exhibiting strong expression and only a few that were faintly expressed (Fig. 2 and Supplementary Fig. 3).”

5. Page 15, lines 20-30 and Figure 4

The authors should indicate the expression hosts for the applied Sx-GTs. The FucT8 appears as a faint band in Figure 2f (FUT8), and this raises the question of whether the E.coli lysate contains an effective FucT8 activity. The authors should assign (ii) Sx- Δ 29HsGnT1 and (iii) Sx- Δ 29HsGnT2 to those lysates shown in Figure 2d.

All Sx-GTs used to generate data in Figure 4 were produced using E. coli BL21(DE3) or its derivative SHuffle T7 Express lysY as host. We have added this information to the Figure 4 caption in our revised manuscript.

Reviewer 1 correctly points out that FucT8 appeared as a fainter band in E. coli lysates as seen in Figure 2; however, simple changes to the growth/induction conditions (e.g., raising the post-induction culture temperature to 30 °C) combined with enrichment afforded by NiNTA purification yielded sufficient material for carrying out the remodeling experiment in Figure 4, which was performed using purified enzymes and not clarified lysates. We have added a Coomassie-stained SDS-PAGE gel showing purification of FucT8 and several additional Sx-GT constructs (see new Supplementary Fig. 6a). As can be seen in this newly added data, there is ample accumulation of FucT8 following its purification from E. coli culture.

GnTI and GnTII are aliases for MGAT1 and MGAT2, respectively. We have revised the labels on Figure 2 to reflect this naming and also to enable assignment of Figure 4 Sx-GTs with corresponding lysates in Figure 2d.

6. Page 17 & 18 and Figure 5

The authors should assign the Sx-GTs depicted in Figure 5a to those lysates shown in Figure 2d.

We have revised the labels in Figure 5a to enable assignment of Figure 5 Sx-GTs with corresponding lysates in Figure 2d.

7. Page 25, lines 3-8:

What was the expression vector for yeast?

Protocol for the transformation of yeast cells is missing.

The vector used for yeast expression was pYS338 as indicated in Supplementary Table 1. We have revised the Methods section to include the information for yeast transformation and expression.

Reviewer #2 (Remarks to the Author):

Jarontomeechia et al. describe how the SIMPLEX approach can be used to produce glycosyltransferases (GTs, incl. many human ones) in E. coli (as well as in yeast and HEK cells). The use of the SIMPLEX approach enables to produce rapidly and cost-effectively functional GTs. Although the SIMPLEX approach has been described before, the current study shows even more convincingly than the previous ones how extremely valuable it is. A subset of the produced Sx-GTs is subsequently successfully used for cell-free remodelling of (free and protein-linked) glycans. This study creates the foundation for a unique and versatile glycoengineering toolkit.

We would like to thank Reviewer 2 for their favorable assessment of our work, in particular pointing out how “the current study shows even more convincingly than the previous ones how extremely valuable [the SIMPLEX approach] is” and how our work “creates the foundation for a unique and versatile glycoengineering toolkit”. In the section that follows, we have addressed the aspects that Reviewer 2 encouraged us to consider.

Minor questions/comments

-In the immuno-blots no whole cell lysates are included. Why?

There was no specific reason that whole cell lysates weren't included; however, we do not feel that these add any additional information that isn't already shown in our immunoblotting analysis. In essence, the soluble, membrane and insoluble fractions together represent the whole cell lysate, thus we felt it was redundant to show the whole cell lysate when these subcellular fractions were already included in all the immunoblots. That said, we have added a new Coomassie stained SDS-PAGE gel of whole cell lysates derived from cells expressing fused and unfused versions of HsST6Gal1, which clearly reveals (i) the strong accumulation of the SIMPLEX construct relative to all other cellular

proteins and (ii) the dramatic expression difference between the Sx- Δ 26HsST6Gal1 fusion relative to its unfused counterpart or versions fused to MBP or ApoAI* only (see newly added Supplementary Fig. 1b).

-Is it possible that the 'Sx fusion partners (MBP and ApoAI)' facilitate the transfer of GTs during immuno-blotting. Have the authors also used Coomassie stained gels to compare the production of proteins with and without the 'Sx fusion partners'?

While it is possible that different proteins can be transferred onto polyvinylidene difluoride (PVDF) membranes with varying efficiencies, we believe these differences are very minor and unlikely to change the immunoblot results in Figure 2 and Supplementary Figure 4. All SDS-PAGE and immunoblot analysis was performed using denatured samples, hence the primary determinants for transfer efficiency are factors related to the primary amino acid sequence such as molecular weight and overall hydrophobicity. It is well known that larger proteins typically transfer to PVDF membranes at a much slower rate than smaller proteins, and longer transfer times (up to 16 h) are often required to enhance the transfer efficiency of larger proteins to the membrane. In our work, we only used standard protein-to-membrane transfer parameters as recommended by the manufacturer (1 hour at constant 20 V), which slightly favors the transfer of smaller proteins (i.e., proteins without the Sx fusion partners). Furthermore, upon inspecting the hydrophobicity of SIMPLEX fusion partners using Protoscale in linear mode employing Kyte and Doolittle parameters, both MBP and ApoAI exhibit average hydrophobicity/hydrophilicity along the protein backbone. Finally, PVDF membranes have been used in qualitative proteomic analyses (for example, Williamson et al, J Microbial Methods 2012; Sebastian et al, MCP 2015) and have demonstrated the ability to capture the proteome in various organisms including bacteria. For all these reasons, we believe the effect of SIMPLEX fusion partners toward transfer efficiency is negligible. That said, we have included a new Coomassie stained SDS-PAGE gel (see Supplementary Figure 1b) that shows whole cell lysates from E. coli cells expressing Sx-ST6Gal1, MBP-ST6Gal1, ST6Gal1-ApoAI, and ST6Gal1. This SDS-PAGE gel analysis provides a clear comparison of the different expression levels of ST6Gal1 with and without the SIMPLEX fusion partners. Importantly, the expression levels observed for whole cell lysates is in near-perfect agreement with our immunoblot results of soluble/insoluble/detergent solubilized fractions and provides further evidence for the ability of the SIMPLEX architecture to promote significantly greater soluble expression compared to unfused proteins lacking the SIMPLEX fusion partners.

-Long term storage conditions are described in the Materials and Methods section: is anything known about how long Sx-GTs can be stored without losing (too much) activity?

We strongly agree with Reviewer 2 that long-term storage is an interesting question. Anecdotally, the Sx-GT enzymes are stable for many months following storage at 4 or -20 °C. However, we have not investigated this in a rigorous fashion and feel that a careful investigation of Sx-GT stability is outside the scope of the current manuscript. Rather, this topic is the focus of a follow-on study in which we are investigating Sx-GT stability in multiple ways (i.e., thermostability, proteolytic stability, storage stability, etc.).

-Can the authors just like for the experiment shown in Figure 4 also comment on the efficiencies of the experiment shown in Figure 5?

Glycan remodeling on trastuzumab was performed in a similar manner as free glycan remodeling with the additional step of protein A/G-based affinity purification between each reaction step. In terms of recovery efficiency, we observed ~80-90% recovery yield of IgG following each purification step. Regarding glycan remodeling efficiency, reactions were nearly 100% efficient following incubation with Sx-GTs for 16-36 h at 37 °C with only the conversion to the disialylated N-glycan using Sx-

Δ26HsST6Gal1 resulting in notably lower efficiency. We have added information about the efficiencies to the Results section of the revised manuscript.

-How pure were the isolated Sx-GTs? In the Materials and Methods section it is mentioned that the purity was assessed using SDS-PAGE followed by Coomassie staining. Can the authors show some representative examples in the supplement?

The purity of isolated Sx-GTs was in the 50-80% range following just a single-step Ni-NTA purification. These purity values were determined by standard densitometry analysis using BioRad Image Lab software, whereby the intensity of the band corresponding to the full-length Sx-GT (indicated with red arrow) was normalized by the intensity of all bands that appeared in the same lane of the gel. To accurately reflect the actual amount of Sx-GTs, reported yields reported in Supplementary Figure 6b were calculated based on both total protein concentration in purified fractions and % purity as estimated from Coomassie-stained gels. We have updated the Methods section of the revised manuscript to more carefully describe how the yields were calculated. Furthermore, as requested by Reviewer 2, we have provided a representative Coomassie-stained gel for three Sx-GTs in newly added Supplementary Figure 6a.

-In Figure 1d and SF6: what is the impact of the MBP and ApoAI moieties? If you have an equal amount of Sx-GT and GT protein, the GT protein should in principle contain more enzymatic activity? Should one somehow correct for the MBP and ApoAI moieties when comparing the activity/expression of Sx-GTs and GTs? Or does this referee totally misses something? In Figure 1d: what represents dark green and what represents light green?

Reviewer 2 correctly points out that an equal mass of Sx-GT will contain fewer molecules (moles) than the same mass of the unfused enzyme and thus should provide less enzymatic activity than an equivalent mass of the non-fused GT. To address this issue, in our revised manuscript we now include careful kinetic analysis for our purified Sx-Δ26HsST6Gal1 enzyme as well as for a commercially sourced version of the unfused enzyme. Specifically, we quantified the apparent kinetic parameters of purified Sx-Δ26HsST6Gal1 and commercial HsST6Ga1 using a conventional activity assay (sialyltransferase activity assay kit; R&D Systems). We determined that the apparent K_M and V_{max} values for our solubilized Sx-Δ26HsST6Gal1 construct were 0.19 ± 0.03 mM and 85.5 ± 6.3 pmol/min, respectively. These values were in close agreement with the apparent kinetic parameters that we measured for the commercial recombinant human ST6Gal1 (R&D Systems; catalog # 7620-GT-010) (see revised Figure 1c) and also in close agreement with measurements made previously by Boons and coworkers (Mbua et al., 2013 Angew Chem Int Ed Engl). The specific activity of the Sx-Δ26HsST6Gal1 enzyme was 516.9 pmol/min/μg, which was consistent with previously published data for the unfused HsST6Gal1 enzyme (Houeix and Cairns, 2019 PeerJ). Therefore, based on these newly added results, we conclude that the MBP and ApoAI domains have little impact on the enzymatic activity of Sx-Δ26HsST6Gal1. These results are now reported in Figure 1c. Moreover, we have added an additional panel to Supplementary Fig. 6 showing a yield comparison for the fused and unfused enzymes on a molar basis. Importantly, the SIMPLEX-reformatted GTs were produced as good or better than the unfused GTs even on a molar basis.

We have also fixed the labeling error in the revised manuscript (now Figure 1c), assigning the appropriate colors to the curves corresponding to each enzyme.

Reviewer #3 (Remarks to the Author):

This manuscript entitled A universal glycoenzyme biosynthesis pipeline that enables efficient cell-free remodeling of glycans by Thapakorn Jaroentomechai et al describes a novel glycosylation remodeling methods using E. coli cell-free system.

Pros:

1. It is a massive piece of work with novelty. The language is appropriate and easy to understand. This novel glycoengineering method indeed has the potential to flexibly and precisely tailor N-glycans on recombinant proteins in cell-free systems.

We thank Reviewer 3 for their positive evaluation of our work and their recognition that our study represents a “massive piece of work with novelty” and the approach “has the potential to flexibly and precisely tailor N-glycans”. In the section that follows, we have addressed the aspects that Reviewer 3 encouraged us to consider.

Cons:

1. The first part about generating soluble Sx-HsST6Gal1 and Sx- Δ 26HsST6Gal1 expressions with functional enzymatic activity verification is convincing and solid. But in Figure 1, miss annotations about the green and orange dots in Figure 1d.

We have fixed this labeling error in the revised manuscript (now Figure 1c), assigning the appropriate colors to the curves corresponding to each enzyme.

2. Then moving forward, the second part for generating massive soluble expression of diverse GTs is impressive but I have questions. Without an enzymatic assay for each Sx-GT, it is hard to believe these engineered GTs originated from prokaryotic and eukaryotic organisms, are functional. Relevant glyco-enzyme activity assay like the one shown in Figure 1 or other methods is necessary. Moreover, any explanation on the low or no expression of some engineered GTs in Figure 2? Figure 2C has a big and continuous overexposed band, please explain. Furthermore, it is recommended to point out if the MW size of each engineered GT is right or not.

We appreciate Reviewer 3's concern about whether the large set of enzymes that we solubilized are functional. Indeed, the most detailed functional characterization was carried out for just a single enzyme, Sx- Δ 26HsST6Gal1, including detailed kinetic analysis of the purified enzyme using a commercial kit (see newly added Fig. 1c). However, the data generated in Figures 4, 5, and Supplementary Figures 11-15 provide direct evidence that an additional 7 enzymes (HsGnTI, HsGnTII, Hs β 4GalT1, HsST3Gal1, HsFucT7, HsFucT8, and CjCstII) are functional within the SIMPLEx architecture based on their ability to chemoenzymatically remodel free or protein-linked glycans. Hence, we feel that relevant glyco-enzyme functional characterization is shown for a reasonable number of Sx-GTs already.

Regarding the low or no expression for some of the Sx-GTs, we do not have a definitive explanation. However, we would like to point out that we only tested one standard growth/induction condition to generate Figure 2 (starting $OD_{600} \approx 0.6$, induction with 0.1 mM β -D-1-thiogalactopyranoside (IPTG) at 16 °C for 16-20 h in LB medium) that was identical for each construct. We did not perform any optimization trials that are commonly associated with expression campaigns employing bacteria. It is our opinion that with a small bit of expression optimization many/all of those faintly expressed constructs could be significantly improved. Indeed, simple changes to the growth/induction conditions (e.g., raising the post-induction culture temperature to 30 °C) combined with enrichment afforded by Ni-NTA purification yielded plenty of material for carrying out the remodeling experiments in Figure 4 and 5. We have added a Coomassie-stained SDS-PAGE gel showing purification of FucT8 and several additional Sx-GT constructs (see new Supplementary Fig. 6a).

Regarding the large, continuous smear for the unfused HsAlg13 enzyme, we do not have a definitive explanation. We suspect that the smear, which is largely below the expected molecular weight for this protein, represents significant degradation of the unfused protein in the E. coli cytoplasm.

Finally, we have carefully double-checked all the MW markers in Figure 2 and they are all correct. In general, all Sx-GT constructs are around 100 kDa (give or take) as a result of appending the MBP and truncated ApoA1 domains. We would like to direct Reviewer 3 to Supplementary Dataset 1 which includes theoretical MWs for all fused and unfused GTs.*

3. This work is with a focus on the cell-free system using E.coli lysate, but it is unclear the applicability of this pipeline in other platforms, as shown in Figure 3, Saccharomyces cerevisiae strain SBY49 and human embryonic kidney (HEK) 293T cells (I assume the authors used SBY49 and HEK cells instead of their lysate) are appropriate for this manuscript or not. It is not clear if SBY49 and HEK in this work are cells or lysates. If those are cell systems, then Figure 3 is somehow irrelevant to the topic of this work. Moreover, in Figure 3, only overexpression of Sx- Δ 26HsST6Gal1 without testing multiple engineered GTs is too early to call that this pipeline is compatible with other systems.

Reviewer 3 is correct that we used cell systems, namely SBY49 yeast cells and HEK293T mammalian cells, instead of their lysates to produce Sx- Δ 26HsST6Gal1. However, we respectfully disagree with Reviewer 3 that Figure 3 is irrelevant to the topic of this manuscript. Rather, we feel that the results presented in Figure 3 are an important facet of the universality/flexibility/versatility of the approach and thus are highly relevant to the manuscript. To this point, we think there may have been some confusion about the nature of our expression workflow and would like to clarify that our study primarily used E. coli cells for expression of all Sx-GTs, whereas functional characterization of Sx-GTs was performed using a variety of in vitro/cell-free glycosylation reaction schemes. E. coli culture was chosen as a proof-of-concept production platform and used to demonstrate successful production of nearly all GTs using a single, standard culture condition as we discussed above. The significance of this achievement relates to the fact that E. coli cell-based expression of GTs, especially of mammalian/human origin, has long been a bottleneck, with most efforts to produce these types of enzymes in bacteria resulting in failure due to poor expression or lack of expression entirely. Importantly, we feel that the novelty of our work stems from the ability of the SIMPLEX technology to promote soluble expression of numerous diverse GTs regardless of their fold-type, origin, or topology. Indeed, this is one important facet of the universality/flexibility/versatility of the SIMPLEX platform.

Another facet of universality/flexibility/versatility that we sought to demonstrate was compatibility across a range of commonly used expression systems which, in addition to E. coli cells, included yeast and mammalian cells as well as E. coli-based cell-free protein synthesis (CFPS). Successful expression of HsST6Gal1 was achieved using all three of these additional expression platforms, which was demonstrated unequivocally in Figure 3 of the manuscript. Hence, the relevance of this figure is related to our claim that the technology provides a universal solution to the production of difficult-to-express GT enzymes. An investigator interested in using our SIMPLEX technology thus has the freedom to express a Sx-GT construct using any of these commonly used expression systems and stands a reasonable chance of generating appreciable amounts of their target GT in a form that is both soluble and functional. Given that we only demonstrated successful expression of HsST6Gal1 in these other platforms, we were careful not to make any claims about the likelihood of successfully producing other GTs in these alternative platforms. Rather, we simply wanted to set the stage for the possibility of using any of the most commonly used expression systems in combination with the SIMPLEX technology, which we believe we have achieved through the inclusion of Figure 3.

Finally, we appreciate Review 3's concern that only one GT (HsST6Gal1) was used to demonstrate compatibility of the SIMPLEX strategy with the various different expression platforms. Indeed, further studies to express SIMPLEX-GTs and their application to prototyping novel glycosylation pathways are underway, but we feel these experiments are beyond the scope of the current manuscript.

4. Mass Spec methods:

Free glycans: Free glycan samples were analyzed by LC-MS/MS. Add time plot or LC plot to the MS plot in Figure 4 to see a complete picture. A full list of glycan identified is necessary as well as these glycan abundances resulting from each glyco-reaction. Moreover, some small peaks in Figure 4b were didn't annotated.

Only free glycans containing sialic acids (glycans 5, 6, 9, and 10) were further analyzed by LC-MS/MS analysis and their LC chromatograms were provided in Supplementary Figures 9a and 10a. All other free glycans were directly analyzed in MALDI-TOF MS analysis after reaction clean-up with graphitized carbon column. To clarify this point, we have more clearly described the specific methods used for preparing and analyzing these glycans in the Method section of our revised manuscript.

Regarding a full list of identified glycans, since we used MALDI-TOF MS to monitor reaction progress until its completion, a full list of glycans is essential for the product generated from each reaction. Such a list was provided in the original manuscript (see Supplementary Table 2). While the disialylation reaction of free-glycans did not go to completion, we have included both mono- and disialylated products in Supplementary Table 2 to accurately reflect this result.

Regarding the small peaks in Figure 4b, we argue that most of these exhibit a mass-to-charge ratio (m/z) less than 500 Da, which is a range that is susceptible to chemical noise that arises from fragmented ions of the matrix and contaminants. As a result, we feel that these peaks should be regarded as background noise and that there is no need to annotate them. Finally, due to low signal-to-noise ratio for MALDI-MS analysis of sialic acid containing glycans, we detected some small peaks within the range m/z 1500-2500 in the MALDI-MS profile. While the identify of these peaks could not be ascertained, we further analyzed sialic acid containing molecules (glycan 5, 6, 9, 10) using LC-MS/MS analysis as shown in Supplementary Figures 10 and 11.

In the method section, it says "For free glycan analysis, the instrument was operated in MS full-scan mode 5 from m/z range from 2,00-2,000 followed by multiple reaction monitoring high-resolution 6 (MRM-HR) scan from 0-12 min at two different collision energies of 20 and 35 V with DP 7 = 20 V and accumulation time of 0.25 s.", while in Figure 4, bi-galactosylated and bi-sialylated N-glycans were identified at M/Z above 2000.

The method noted by Reviewer 3 relates to how we analyzed free-glycans containing sialic acids and corresponds to the results shown in Supplementary Figures 9 and 10. For analyzing glycans 5, 6, 9, and 10, peaks corresponding to the $(M+2H)^{2+}$ mass were selected for fragmentation analysis. The mass-to-charge ratio (m/z) of the $(M+2H)^{2+}$ ion and its fragments fall within 200-2000 Da. The spectrum in Figure 4 was generated using MALDI-TOF MS analysis, which was not clearly indicated in the original manuscript. To clarify this point, we have added details on MALDI-TOF MS analysis to the Method section of the revised manuscript.

5. The significantly different band intensity of internal controls on WB blots: Supplemental Figure 1 and supplemental figure 3.

We appreciate Review 3's concern that a few internal control blots in Supplementary Figures 1 and 3 contain different band intensities. However, we believe that these differences are insignificant and unlikely to materially change the conclusions drawn from these results. Furthermore, as immunoblot analysis is semi-quantitative at best, we believe control blots in Supplementary Figures 1 and 3 provide reasonable assurance that similar amounts of protein were loaded in the SDS-PAGE and immunoblot analyses.

REVIEWERS' COMMENTS

Reviewer #1 (Remarks to the Author):

The authors addressed all critical points and included appropriate corrections.

Reviewer #2 (Remarks to the Author):

No further questions/comments.